EMBO
Molecular Medicine

# An autocrine ActivinB mechanism drives TGFβ/Activin signaling in Group 3 medulloblastoma

Morgane Morabito[1,2,3,4,5], Magalie Larcher[1,2,3,4,5], Florence MG Cavalli[6,7], Chloé Foray[1,2,3,4,5], Antoine Forget[1,2,3,4,5], Liliana Mirabal-Ortega[1,2,3,4,5], Mamy Andrianteranagna[5,8,9,10,11,12,13], Sabine Druillennec[1,2,3,4,5], Alexandra Garancher[1,2,3,4,5], Julien Masliah-Planchon[5,8,9,11], Sophie Leboucher[1,4], Abel Debalkew[6,7], Alessandro Raso[14], Olivier Delattre[5,8,9,11], Stéphanie Puget[15,16], François Doz[8,11,15], Michael D Taylor[6,7,17,18], Olivier Ayrault[1,2,3,4,5], Franck Bourdeaut[5,8,9,10,11], Alain Eychène[1,2,3,4,5] (ID) & Celio Pouponnot[1,2,3,4,5,*] (ID)

## Abstract

Medulloblastoma (MB) is a pediatric tumor of the cerebellum divided into four groups. Group 3 is of bad prognosis and remains poorly characterized. While the current treatment involving surgery, radiotherapy, and chemotherapy often fails, no alternative therapy is yet available. Few recurrent genomic alterations that can be therapeutically targeted have been identified. Amplifications of receptors of the TGFβ/Activin pathway occur at very low frequency in Group 3 MB. However, neither their functional relevance nor activation of the downstream signaling pathway has been studied. We showed that this pathway is activated in Group 3 MB with some samples showing a very strong activation. Beside genetic alterations, we demonstrated that an ActivinB autocrine stimulation is responsible for pathway activation in a subset of Group 3 MB characterized by high PMEPA1 levels. Importantly, Galunisertib, a kinase inhibitor of the cognate receptors currently tested in clinical trials for Glioblastoma patients, showed efficacy on orthotopically grafted MB-PDX. Our data demonstrate that the TGFβ/Activin pathway is active in a subset of Group 3 MB and can be therapeutically targeted.

**Keywords** activin; medulloblastoma; Smad2; Smad3; TGFbeta
**Subject Category** Cancer

## Introduction

Medulloblastoma (MB), a cerebellar tumor, is one of the most common malignant brain tumors in children (Holgado *et al*, 2017; Wang *et al*, 2018). Current therapy associates surgery, chemotherapy, and radiotherapy. This aggressive regimen allowed an increase in the overall survival rate up to 70–80% but induces dramatic long-term side effects (Martin *et al*, 2014). In addition, the overall survival rate of high-risk patients is far below (Holgado *et al*, 2017; Wang *et al*, 2018). It is therefore crucial to identify new treatments that decrease side effects and improve efficacy.

Genomic and transcriptomic approaches allowed the stratification of MB patients into 4 different molecular groups: WNT (Wingless), SHH (Sonic Hedgehog), Group 3, and Group 4 (Northcott *et al*, 2012a; Taylor *et al*, 2012). These groups display differences in

1  Institut Curie, Orsay, France
2  INSERM U1021, Centre Universitaire, Orsay, France
3  CNRS UMR 3347, Centre Universitaire, Orsay, France
4  University Paris Sud − Paris-Saclay, Orsay, France
5  PSL Research University, Paris, France
6  The Arthur and Sonia Labatt Brain Tumour Research Center, The Hospital for Sick Children, Toronto, ON, Canada
7  Developmental and Stem Cell Biology Program, The Hospital for Sick Children, Toronto, ON, Canada
8  Institut Curie, Paris, France
9  INSERM U830, Paris, France
10 Translational Research in Pediatric Oncology, Institut Curie SiRIC, Paris, France
11 SIREDO Center (Care, innovation, Research in pediatric, adolescent and young adult oncology), Institut Curie, Paris, France
12 INSERM, U900, Paris, France
13 MINES ParisTech, CBIO-Centre for Computational Biology, Paris, France
14 Department of Patology, ASL 3 Genovese, SC Laboratorio d'Analisi, Genova, Italy
15 Université Paris Descartes, Sorbonne Paris Cité, Paris, France
16 Département Neurochirurgie Pédiatrique, AP-HP, Hôpital Necker-Enfants Malades, Paris, France
17 Department of Laboratory Medicine and Pathobiology, University of Toronto, Toronto, ON, Canada
18 Division of Neurosurgery, The Hospital for Sick Children, Toronto, ON, Canada
*Corresponding author. Tel: +33 1 69 86 30 79; Fax: +33 1 69 86 30 51; E-mail: celio.pouponnot@curie.fr

terms of cell of origin, transcriptional, epigenetic, and mutational signatures. They also differ in their clinical characteristics such as histology, overall survival rate, and presence of metastases. Recently, intragroup heterogeneity has been further uncovered, allowing their division into subtypes with some specific clinical parameters as well as genomic alterations (Cavalli et al, 2017a; Northcott et al, 2017; Schwalbe et al, 2017). Although the existence of subdivisions within the different groups is clear, the outlines of the different subtypes have not completely reached a consensus so far. The WNT group represents 10% of all MBs and is driven by constitutive activation of the WNT/β-catenin pathway with patients showing the best prognosis. The SHH group accounts for 20–25% of MB and is characterized by mutations involving different mediators of the SHH pathway. It is considered of intermediate prognosis. However, recent sub-classifications identified SHH subtypes with poorer outcomes (Cavalli et al, 2017a; Schwalbe et al, 2017). On the other side, Group 3 and Group 4 are far less characterized due to their genetic and clinical heterogeneity. They display some degrees of overlap with a few samples (~10%) being difficult to specifically assign to either Group. They share some clinical characteristics, such as a high propensity to metastasis and genetic alterations such as OTX2 amplifications or KBTBD4 mutations (Northcott et al, 2017). In contrast to SHH and WNT groups, no deregulation of a given signaling pathway has been yet reported. Group 4 represents 35–40% of all MB patients and shows, in few cases, MYCN and CDK6 amplifications and KDM6A mutations. Recently, it has been shown that genomic alterations involving enhancer hijacking induce PRDM6 overexpression in 15–20% of Group 4 (Northcott et al, 2017). Group 3 represents 20–25% of MB patients and is associated with bad prognosis. This group is highly metastatic and characterized by MYC overexpression, which can be explained in 15–20% of cases by its amplification. However, MYC overexpression is not sufficient to induce Group 3 MB and requires additional cooperating oncogenic events (Kawauchi et al, 2012; Pei et al, 2012). Some of them have been identified, such as GFI1 and GFI1B that are highly expressed in a subset of Group 3 through enhancer hijacking (Northcott et al, 2014). These transcription factors have been demonstrated to drive Group 3 MB tumorigenesis in animal models when associated with MYC overexpression (Northcott et al, 2014). At the transcriptomic level, Group 3 is characterized by the expression of a photoreceptor program defined by genes whose expression is highly restricted to the retina (Kool et al, 2008; Cho et al, 2011). We recently uncovered that this program defines a subtype within Group 3 tumors, which exhibits a functional dependency to this ectopic program through its two main drivers, the retina-specific transcription factors NRL and CRX (Garancher et al, 2018). Thus, Group 3 can be subdivided into 2–3 different subtypes according to the different studies (Cavalli et al, 2017a; Northcott et al, 2017; Schwalbe et al, 2017). Cavalli et al (2017a) have identified 3 subtypes, one is composed of tumors with high MYC expression including those with amplification of this gene, named G3γ. This subtype has the worse prognosis. The second subtype, G3β, is over-represented by tumors with GFI1 alterations, and the last one G3α, by tumors expressing photoreceptor genes in which few amplifications of mediators of the TGFβ/Activin pathway can be found (Cavalli et al, 2017a). Since Group 3 displays the worse prognosis, targeted therapies are actively searched. Different actionable targets have been proposed mainly based on genomic data, including the

TGFβ signaling, which has been suggested to be deregulated in few Group 3 MB, although no functional data have been reported so far. A study on structural genomic variations across over 1,000 MB has first described few amplifications of different mediators of the TGFβ/Activin pathway in Group 3 MB (Northcott et al, 2012b). They include ACVR2A and ACVR2B, two type II receptors for Activin, as well as TGFBR1, a type I receptor for TGFβ, highlighting a potential deregulation of Smad2/3 signaling (see below). Additionally, since OTX2 has been demonstrated to be a target gene of this signaling pathway (Jia et al, 2009), it has been proposed that OTX2 amplifications could represent a mechanism by which the pathway is also deregulated downstream (Northcott et al, 2012b). The putative significance of this signaling pathway in Group 3 was reinforced by two subsequent studies, one involving sequencing in a large cohort of MB (Northcott et al, 2017) and the other showing that several components of this signaling pathway could also be deregulated at their expression level, through Group 3-specific enhancers (Lin et al, 2016). Although these studies might indicate a potential deregulation of the Smad2/3 signaling pathway, this could account for only a modest proportion of Group 3 tumors.

The TGFβ superfamily is a large family of cytokines divided into two distinct groups of ligands: the TGFβs/Activins and the BMPs. TGFβ/Activin ligands signal through Smad2/3. These ligands bring together two types of serine/threonine kinase receptors, the type I and the type II, which are specific for a set of ligands. The TGFβs (TGFβ1, TGFβ2, and TGFβ3) signal through the TGFBR1 type I and TGFBR2 type II receptors. Activin, encoded by 4 different genes, INHBA, INHBB, INHBC, and INHBE, can activate different couples of receptors including the ACVR2A and ACVR2B type II and ACVR1A (ALK4) and ACVR1C (ALK7) type I receptors. INHA, encoding inhibin-α, is an inhibitor of the Activin ligands. Activin and TGFβ ligands lead to the phosphorylation and activation of the same intracellular mediators, Smad2 and Smad3, which then associate with the co-Smad, Smad4. The hetero-complex translocates to the nucleus, where it activates the transcription of target genes with the help of DNA binding partners (Levy & Hill, 2006; Ross & Hill, 2008).

TGFβ/Activin signaling displays pleiotropic functions depending on the cellular and environmental context. Its implication in cancer has been well documented, mainly through TGFβ ligands, although BMPs and Activins ligands can be also involved (Seoane & Gomis, 2017). The role of the TGFβ signaling pathway in cancer is complex, acting either as a tumor suppressor pathway in some instances or as a tumor promoter in others (Massagué, 2008; Seoane & Gomis, 2017). Its oncogenic role is mainly associated with an autocrine (or paracrine) stimulation, due to the strong expression of TGFβ ligands. The TGFβ pathway has been shown to promote cell proliferation in specific context such as in Glioblastoma (Bruna et al, 2007) and cancer stem cell maintenance (Peñuelas et al, 2009; Anido et al, 2010; Lonardo et al, 2011). Studies on the role of Activin ligands in cancer are much more scarce (Wakefield & Hill, 2013). By activating the same mediators Smad2/3, a parallel can be drawn between TGFβ and Activin. Indeed, Activins act both as tumor suppressors and tumor promoters (Chen et al, 2002; Antsiferova & Werner, 2012; Marino et al, 2013; Wakefield & Hill, 2013). Their pro-tumorigenic role has been validated in animal models in which deletion of the activin inhibitor, INHA, led to gonadal tumors in mice as well as cachexia-like syndrome (Matzuk et al, 1994; Vassalli et al, 1994). ActivinB has also been shown to play a role in

cancer stem cell maintenance (Lonardo et al, 2011) and in cell dedifferentiation in an insulinoma mouse model and deletion of *INHBB* encoding ActivinB increases survival (Ripoche et al, 2015).

Several observations pinpoint to a potential role of the Smad2/3 signaling pathway in Group 3 MB but no published data have confirmed the deregulation of this signaling pathway, nor its functional involvement in Group 3 biology. In this study, we investigated these aspects to bring the proof of principle that this signaling pathway represents an interesting therapeutic target in MB and to identify patients that could be eligible to such therapy.

## Results

### TGFβ/ActivinB signaling pathway is active in Group 3 MB

Since different genomic alterations in the TGFβ/Activin pathway have been previously described in Group 3 MB (Northcott et al, 2012b, 2017; Lin et al, 2016), we first investigated whether the pathway is activated in patient samples. We performed WB analysis on 38 medulloblastomas: 7 WNT, 12 SHH, 10 Group 3, and 9 Group 4 tumors. Activation of the pathway, monitored by the level of Smad2 phosphorylation (P-Smad2), was observed in some patient samples from all MB groups (Fig 1A). An inter-tumor heterogeneity was observed in each group, with some samples with high P-Smad2. However, an overall higher level of Smad2 phosphorylation was observed in Group 3 when normalized to β-actin (Fig EV1A, left panel). This was not evidenced when normalized to total Smad2 (Fig EV1A, right panel) since an important variation of Smad2 level was observed (Fig 1A). This is in line with the modification of Smad2 stability by auto-regulatory mechanisms (Yan et al, 2018). Thus, the overall level of P-Smad2/β-actin, which formally reflects the level of nuclear and active Smad2, led us to conclude that TGFβ/Activin pathway is activated in some Group 3 patients.

Considering that amplifications of receptors of the pathway have been described in less than 10% of Group 3 tumors (Northcott et al, 2012b), we hypothesized that other mechanisms may account for pathway activation in several G3 samples. Activation of the Smad2/3 pathway in cancer is frequently due to autocrine/paracrine activation by TGFβ ligands (Rodón et al, 2014). Therefore, we analyzed the expression of major mediators of the TGFβ/Activin pathway, including ligands and receptors in previously published MB dataset at the mRNA (Data ref: Cavalli et al, 2017b) and protein (Data ref: Archer et al, 2018b) levels. No major difference in the expression of the different receptors was observed between the different groups (Fig EV1B and C). In contrast, striking differences were observed for the ligands. For example, *TGFB2* was found highly expressed in SHH tumors (Fig EV1B and C and Appendix Table S1). We observed higher expression levels of *TGFB1*, *TGFB3*, and *INHBB* (encoding ActivinB) in Group 3 in comparison with the other ones although expression of *TGFB3* is similar between Group 3 and Group 4 (Fig 1B). These results were confirmed at the protein level (Fig EV1C). These data were compatible with an autocrine activation of the pathway by one of those ligands listed above in Group 3 MB.

We next investigated the activation of TGFβ/Activin pathway in MB cell lines. We analyzed the level of P-Smad2 in four well-established Group 3 MB cell lines (HDMB03, D458, 1603MED, and D283)

as well as in three cell lines classified as non-Group 3 (DAOY, ONS76, and UW228). Western Blot (WB) analyses showed higher basal intensity of P-Smad2 signal in Group 3 cell lines (Fig 1C), confirming that the pathway is activated in this group. As in patient samples, we observed heterogeneity in the activation of the pathway, with a very strong basal level of pathway activation being observed in the 1603MED cell line while in some cell lines its level was modest.

To understand what drives the basal activation of the pathway in Group 3 cell lines, we investigated the expression level of different ligands and receptors of the pathway by RT–qPCR (Figs 1D and EV1D). No marked difference in the expression of the receptors was found between Group 3 and non-Group 3 cell lines, except a higher expression of *ACVR1B*, *ACVR2A*, and *ACVR2B* (Fig EV1D and Appendix Table S2) and a lower expression of *TGFBR2*, an obligatory partner for *TGFBR1*, in Group 3 cell lines. We did not observe any direct correspondence between the expression of the different receptors and the level of activation of the pathway in the different Group 3 cell lines (i.e., level of P-Smad2 in Fig 1C), suggesting that pathway activation is not directly linked to the deregulation of receptors expression. We investigated the expression of different ligands (Figs 1D and EV1D) and found a higher expression of *INHBB* in the 1603MED and D283 Group 3 cell lines as compared to the others (Fig 1D and Appendix Table S2). Interestingly, this level of expression directly corresponded to that of P-Smad2 levels, strong in 1603MED to intermediate in D283. This suggested that the ActivinB, encoded by *INHBB*, could be the major driver of Smad2/Smad3 phosphorylation in this group. The same observation could be drawn for *TGFB3* in 1603MED and to a lesser extent in D283, while genes encoding the other ligands were not overexpressed in the cell lines showing a high level of P-Smad2 (Fig EV1D). Taken together, these results suggested the potential existence of an autocrine mechanism involving either *TGFB3* or *INHBB* that could be responsible for TGFβ/Activin signaling activation in Group 3 MB.

### An autocrine stimulation involving ActivinB

To further investigate the presence of a potential autocrine mechanism, we first analyzed the ability of cell lines to respond to exogenous stimulation by either TGFβ or Activin ligands, each requiring different sets of receptors. Non-Group 3 cell lines showed an increase in P-Smad2 signals in response to TGFβ stimulation, while no modulation was observed upon Activin stimulation (Fig 2A, left in blue). Strikingly, Group 3 MB cell lines showed the complete opposite profile: P-Smad2 signal was increased upon Activin stimulation, while it remained unchanged upon TGFβ stimulation (Fig 2A, right in yellow). Noteworthily, 1603MED displayed a very high basal level of P-Smad2 which is constitutive. The reason for which G3 cell lines respond to Activin but not to TGFβ is currently unknown. However, we noticed a lower level of *TGFBR2* in these cells, a receptor required for TGFβ response (Fig EV1D). This was also observed in G3 tumor samples at the RNA and protein level (Fig EV1B and C). These opposite responses suggested a ligand-specific response between MB subgroups with Group 3 MB cell lines being able to respond to Activin but not to TGFβ, thereby excluding TGFβ ligands as a potential autocrine source for Smad2 activation. Since Group 3 cell lines displayed concomitant pathway activation and *INHBB* expression, these results strongly suggested that an

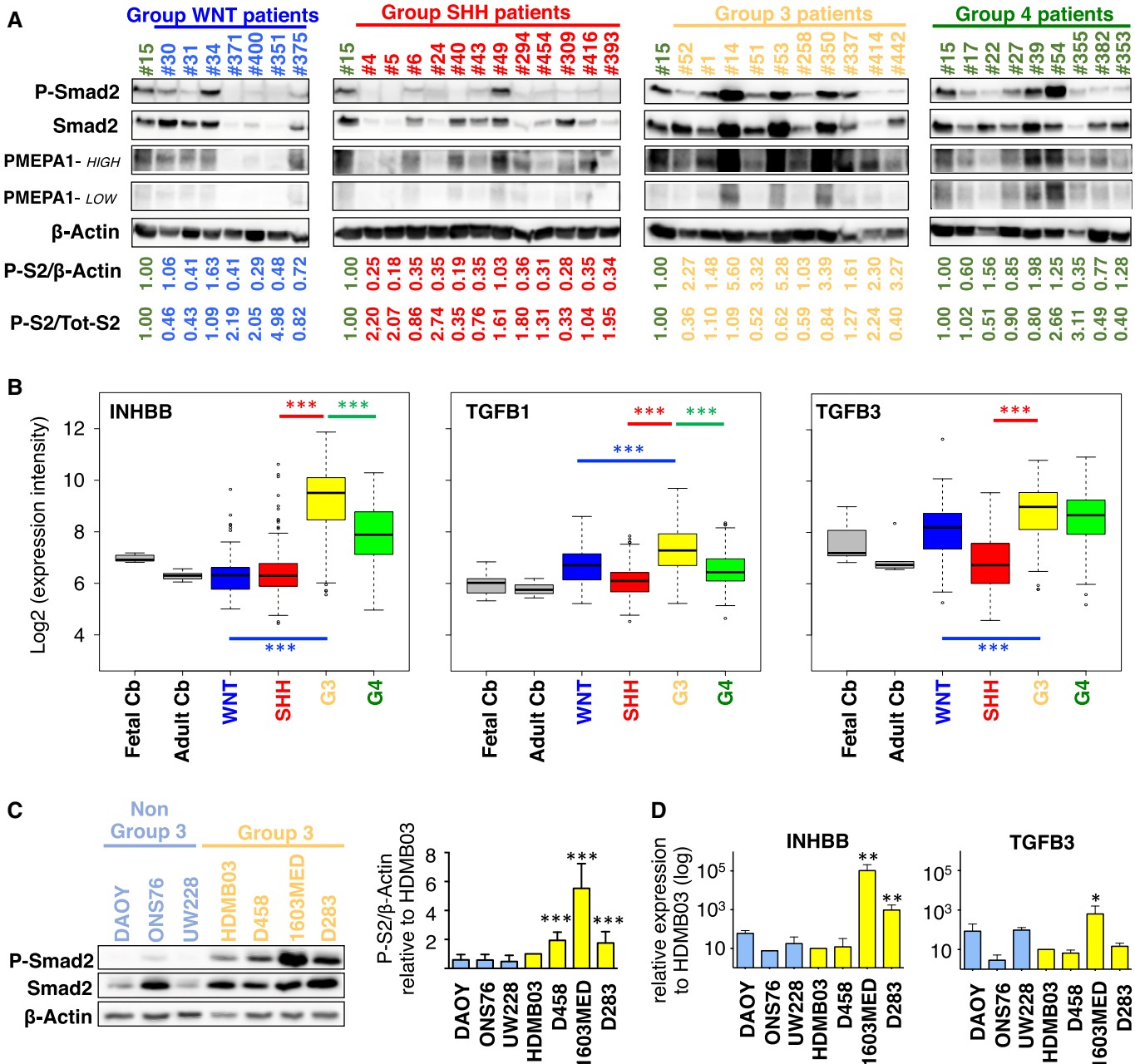

**Figure 1. TGFβ/ActivinB pathway is activated in Group 3 MB patients and cell lines.**

A  Immunoblot analysis of phosphorylated Smad2 (P-Smad2) and PMEPA1 (high and low exposures displayed) in MB patient sample lysates from different groups: WNT (blue), SHH (red), Group 3 (yellow), or Group 4 (green). β-Actin was used as a loading control. Relative quantification of P-Smad2 signal to β-actin (P-S2/β-Actin) and total Smad2 (P-S2/Tot-S2) are indicated below the blots.

B  Boxplots summarizing the expression of *INHBB*, *TGFB1*, and *TGFB3* ligands of the TGFβ/Activin pathway in the different groups of MB (blue WNT, red SHH, yellow Group 3, and green Group 4) and in fetal and adult cerebellum (gray) in the dataset of Cavalli *et al* (Data ref: Cavalli *et al*, 2017b).

C  Immunoblot analysis of phosphorylated Smad2 (P-Smad2) in non-Group 3 (blue) and Group 3 (yellow) MB cell lines on the left panel. The level of total Smad2 (Smad2) was assessed, and β-actin was used as a loading control. On the right panel, relative level of P-Smad2 (P-S2) was quantified to total β-actin. P-Smad2 to total Smad normalization is also provided on Appendix Fig S5.

D  RT–qPCR was performed on RNA extracted from non-Group 3 (blue) and Group 3 (yellow) MB cell lines to compare expression levels of *INHBB* (left) and *TGFB3* (right).

Data information: Wilcoxon rank-sum tests were performed to determine *P*-values for panel (B). Boxplot center lines show data median; box limits indicate the 25th and 75th percentiles; lower and upper whiskers extend 1.5 times the interquartile range (IQR) from the 25th and 75th percentiles, respectively. Outliers are represented by individual points (B). The remaining *P*-values were determined by unpaired *t*-test. *$P < 0.05$, **$P < 0.01$, ***$P < 0.001$, ****$P < 0.0001$. Bars represent the mean ± SD. Number of replicates is $n \geq 3$. The exact *P*-values and number of replicates are indicated in Appendix Table S5. Detailed statistics are presented in Appendix Table S1 for panel (B) and Appendix Table S2 for panel (D).

Source data are available online for this figure.

ActivinB (encoded by *INHBB*) autocrine stimulation could be responsible for the activation of the pathway in the 1603MED and D283 Group 3 cell lines. To further investigate the potential role of ActivinB in the basal Smad2 activation in these cell lines, we focused on the 1603MED cell line, which shows the strongest basal activation. Treatment of 1603MED cells with an ActivinB blocking antibody induced a decrease in P-Smad2 level (Fig 2B). Importantly, the specificity of this antibody toward ActivinB was verified by showing that it does not block TGFβ stimulation (Appendix Fig S1). These experiments supported that an autocrine ActivinB production induced, at least partially, a strong activation of the pathway in 1603MED. This was further supported by P-Smad2 inhibition upon treatment with follistatin, a ligand trap for Activins (Fig 2B). We next sought to directly demonstrate that 1603MED cells secrete ActivinB. HDMB03 cells were used as receiving cells to conditioned media, since they showed the lowest basal activation of the pathway among G3 cell lines (Fig 1C) but efficiently responded to exogenous ActivinB and not to TGFβ stimulation (Fig 2A). Three culture media were tested as follows: a non-conditioned media that had never been in contact with any cells, an HDMB03-conditioned media, both of them being used as negative controls, and a 1603MED conditioned media. HDMB03 cells were treated with these different media for 1 h, and the effect on the Smad2 pathway was tested by WB (Fig 2C). 1603MED conditioned media induced a strong P-Smad2 signal as compared to the two control media. This induction was prevented by incubation with an ActivinB blocking antibody (Fig 2C), strongly supporting that 1603MED secreted active ActivinB ligand. To further substantiate this hypothesis, we targeted INHBB expression by siRNA in 1603MED. Although expression of *INHBB* was reduced to only 40% (Fig 2D), we nonetheless observed a decrease in P-Smad2 level (Fig 2E) resulting in decreased cell growth (Fig 2F). All these effects were rescued by exogenous addition of ActivinB (Fig EV2). Altogether, these results strongly support an autocrine secretion of ActivinB by 1603MED cells leading to P-Smad2 activation and promoting 1603MED cell proliferation.

### ActivinB stimulation promotes proliferation

We next investigated the role of Activin pathway activation in Group 3 MB cell lines. D458 (Fig 3A–D) and D283 (Fig 3E–H) cells, which showed intermediate basal activation of the pathway (Fig 1C), were stimulated with ActivinB (Fig 3). Activation of the pathway was validated by monitoring P-Smad2 levels (Fig 3A and E). Incucyte Proliferation Assay revealed an increase in cell proliferation upon ActivinB stimulation in both cell lines (Fig 3B and F). It remains to be determined why ActivinB did not promote cell growth while activating the pathway in HDMB03 (Appendix Fig S2). An increase in cell proliferation can result from faster cell cycle progression, a reduction in cell death, or both. We analyzed the cell cycle profile by BrdU incorporation and 7AAD labeling and observed an increase in the number of cells in S phase following ActivinB stimulation, concomitant with a decrease in G0/G1 (Fig 3C and G). Apoptosis was monitored by FACS analysis of cleaved caspase-3 staining. We did not detect consistent effects on apoptosis, with a slight decrease in D458 cell line following stimulation after 2 days (Fig 3D), while no changes were detected in D283 (Fig 3H). These results indicated that ActivinB stimulates cell proliferation in Group 3 cell lines mainly by promoting cell cycle progression.

### Inhibition of the pathway decreases proliferation

We next investigated the consequences of pharmacological inhibition of the pathway in Group 3 MB cell lines (Fig 4). One Group 3 cell line that exhibits a very high basal activation of the pathway (1603MED, Fig 4A–D) and one with an intermediate level (D283, Fig 4E–H) were treated with LY364947 or SB431542. These compounds prevent the phosphorylation of Smad2/3 by the TGFβ and Activin type I receptors. Indeed, we verified that they prevent TGFβ- as well as ActivinB-induced P-Smad2 (Appendix Fig S1). After 24 h of treatment, the level of P-Smad2 was decreased in 1603MED and D283 cell lines (Fig 4A and E, respectively). This pathway inactivation was accompanied by a decrease in cell proliferation (Fig 4B and F). FACS analyses were performed to measure BrdU incorporation and 7AAD labeling. Treatment with inhibitors induced a decrease in the percentage of cells in S phase concomitant with an increase in G0/G1 (Fig 4C and G). A very slight increase in the percentage of cells positive for cleaved caspase-3 staining was also observed (Fig 4D and H), showing that the inhibition of the pathway mainly impacted on cell cycle and to a much lesser extent on apoptosis.

### PMEPA1 is implicated in ActivinB promotion of cell growth

To identify relevant genes downstream of Activin signaling in Group 3 MB, we sorted the top 10 genes, whose expression was correlated with *INHBB* in Group 3 patient samples (Fig 5A). *PMEPA1*, which scored as the top gene, is a well-established Smad2/3 target gene in different cell types including P19 cells stimulated by Activin (Coda *et al*, 2017). Accordingly, we found that *PMEPA1* expression level was enriched in Group 3 MB (Fig 5B) and correlated with *INHBB* expression in MB (Fig 5C). This correlation is highest in G3 as compared to the other groups (Appendix Fig S3A). Accordingly, we observed a good correspondence between P-Smad2 overall level and PMEPA1 protein expression in patient samples by Western blot analysis (Figs 1A and 5D and E, Appendix Fig S3B). We next tested whether *PMEPA1* is also a target of the Smad2 signaling in MB by modulating pathway activation (Fig 5F). Activation of the pathway by ActivinB induced an increase in *PMEPA1* mRNA and protein levels, while its inhibition by LY364947, SB431542, blocking ActivinB antibody, or follistatin had the opposite effect in G3 cell lines (Fig 5F and Appendix Fig S3C and D). *MYC* and *OTX2* are key players in Group 3 MB and are also known as Smad2/3 target genes in other cell types (Jia *et al*, 2009; Brown *et al*, 2011; Coda *et al*, 2017). Therefore, we investigated whether their expression could be modulated by this pathway in Group 3 MB cell lines. In contrast to *PMEPA1*, no major change was observed at the mRNA (Appendix Fig S3C) and protein (Fig 5F and Appendix Fig S3D) levels upon pathway inhibition regarding *OTX2*, while a slight decrease could be observed for *MYC*. However, no significant increase in *MYC* expression was observed upon ActivinB treatment (Appendix Fig S3C). Interestingly siRNA-mediated INHBB knockdown decreased PMEPA1 expression that could be rescued upon ActivinB treatment (Fig EV3A). These results suggested that *PMEPA1* is a target gene of the Activin pathway in Group 3 MB but that neither *MYC* nor *OTX2*, two important players of this group, appears to be consistently regulated by this signaling pathway although minor effects are observed on *MYC*. The role of PMEPA1 in cancer remains unclear and is likely to be cell type specific. It has been shown to either promote or

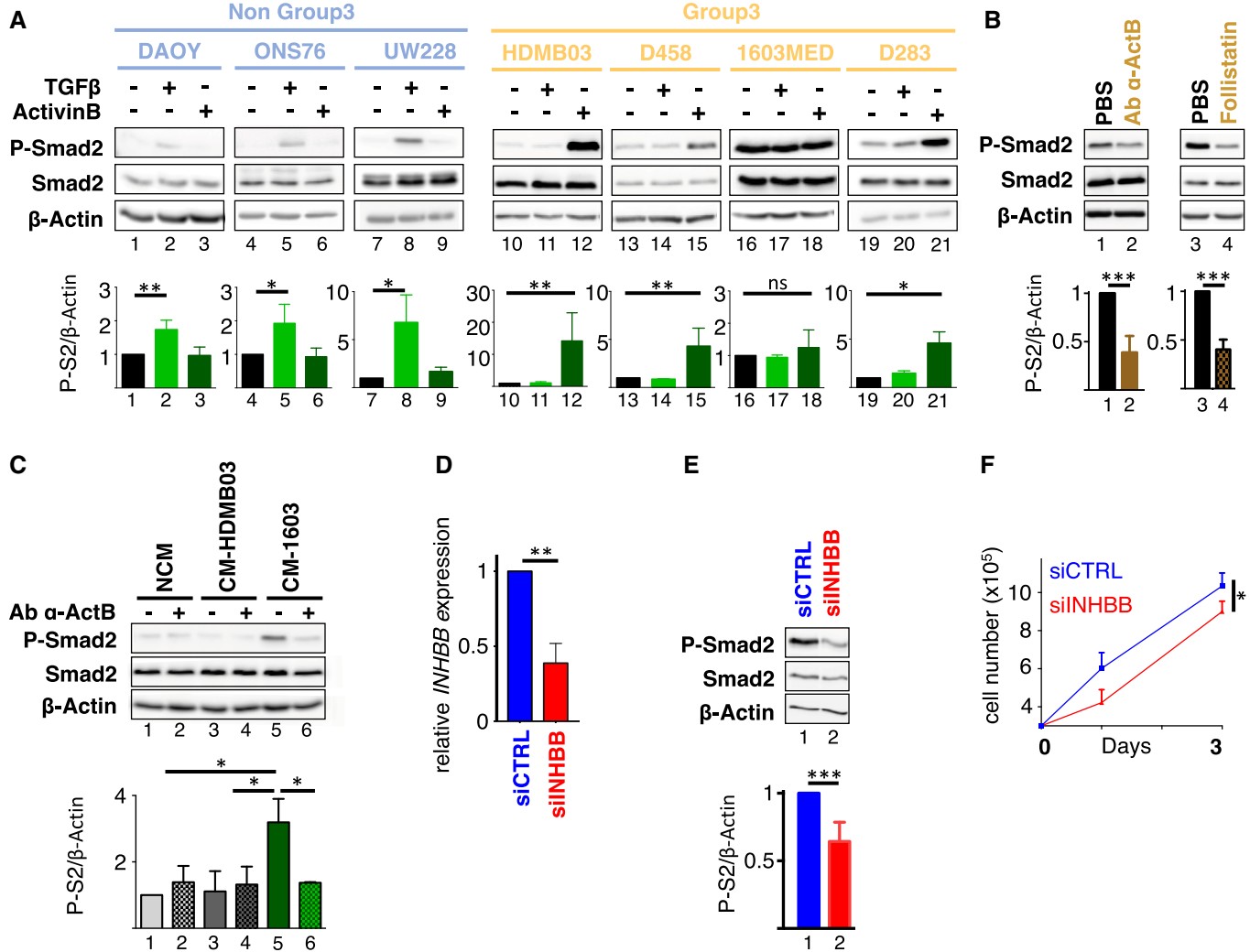

**Figure 2. An autocrine stimulation by ActivinB in the 1603MED cell line.**

A–C The level of phosphorylated Smad2 (P-Smad2) and total Smad2 (Smad2) was assessed by immunoblotting, and β-actin was used as a loading control. Lower bar graphs show WB quantification of P-Smad2 (P-S2) normalized to β-actin. (A) Activation of the pathway was assessed in non-Group 3 (blue) and Group 3 (yellow) MB cell lines in response to TGFβ or ActivinB stimulation for 1 h. (B) 1603MED cells were treated with PBS (vehicle) or a blocking antibody targeting ActivinB (Ab α-ActB) or follistatin. (C) Conditioned media experiments were performed on the HDMB03 MB cell line. Phosphorylation of Smad2 was analyzed by immunoblot upon treatment with either non-conditioned media (NCM), media conditioned with HDMB03 cells (CM-HDMB03), or media conditioned with 1603MED cells (CM-1603). Pre-incubation with blocking antibody against ActivinB (Ab α-ActB) or vehicle (PBS) was performed before HDMB03 cell-line treatment as indicated. Relative level of P-Smad2 (P-S2) was quantified to β-actin (below).

D RT–qPCR was performed on total RNA extracted from 1603MED cells 48 h after transfection with siRNA targeting INHBB. Relative *INHBB* expression was assessed. siCTRL condition was set at 1.

E 1603MED cells were transfected with the indicated control siRNA (siCTRL, blue) or targeting INHBB (siINHBB, red). Lysates were prepared 48 h after transfection. The level of phosphorylated Smad2 (P-Smad2) and total Smad2 (Smad2) was assessed by immunoblot, and β-actin was used as a loading control. Lower bar graphs represent the quantification of the relative level of P-Smad2 (P-S2) to β-actin.

F Growth curve of 1603MED cells after transfection with either siCTRL (blue) or siINHBB (red).

Data information: P-Smad2 to total Smad normalization is provided on Appendix Fig S5. The *P*-values were determined by unpaired *t*-test. *$P < 0.05$, **$P < 0.01$, ***$P < 0.001$. Bars represent the mean ± SD. Number of replicates is $n \geq 3$. The exact *P*-values and number of replicates are indicated in Appendix Table S5. Source data are available online for this figure.

restrain cancer progression (Liu *et al*, 2011; Fournier *et al*, 2015; Nie *et al*, 2016). Therefore, we investigated its role in Group 3 MB. siRNA-mediated PMEPA1 knockdown resulted in cell growth inhibition in both 1603MED and D283 cell lines (Figs 5G–J and EV3B–E), suggesting that PMEPA1 is an important mediator of Activin signaling-mediated proliferation in Group 3 MB.

**TGFβ/ActivinB signaling pathway in Group 3 MB Patient Derived Xenografts (PDXs)**

We further validated the importance of the pathway in patient derived xenograft (PDX) models, known to remain close to the original tumor (Fig 6). As observed in Group 3 patient samples and cell

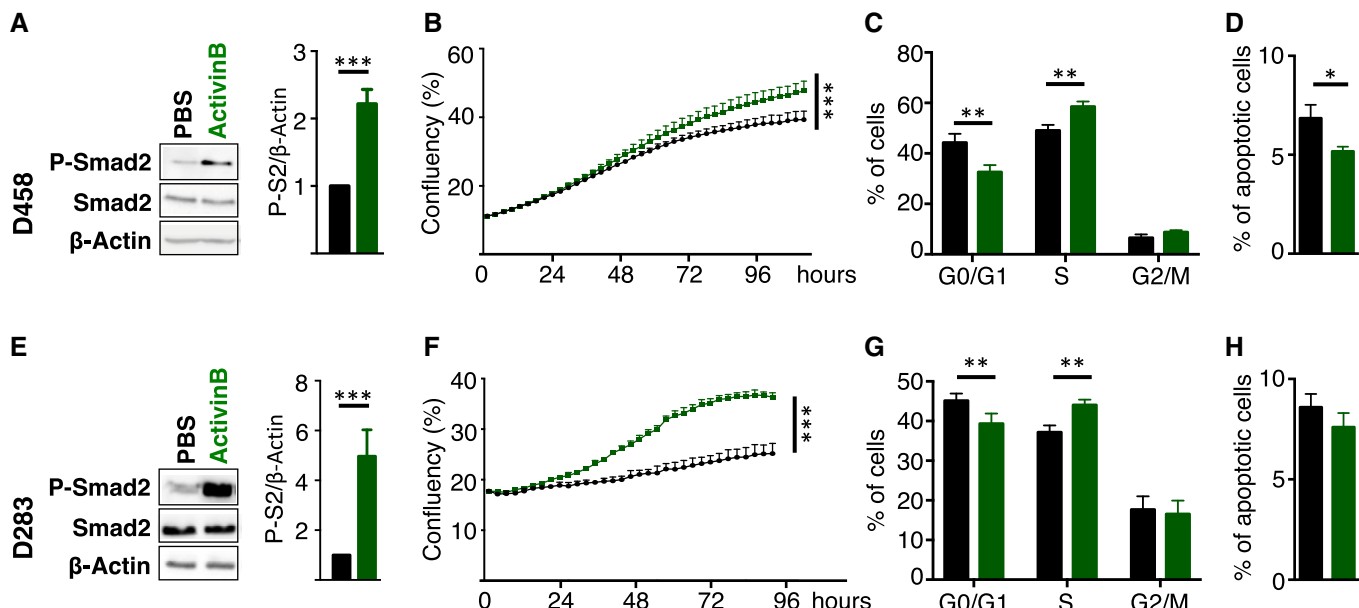

**Figure 3. ActivinB promotes cell proliferation in Group 3 MB cell lines.**

A–H   D458 (A–D) or D283 (E–H) cell lines were treated with PBS (vehicle, black) or with ActivinB (green). (A and E) Immunoblot of phosphorylated Smad2 (P-Smad2), total Smad2, and β-actin in response to ActivinB stimulation for 24 h. Quantification of P-Smad2 (P-S2) to β-actin is shown on right panels. (B and F) P-Smad2 to total Smad normalization is provided on Appendix Fig S5. Growth curve experiments showing cell proliferation upon ActivinB treatment. (C and G) Cell cycle analysis by FACS measuring BrdU incorporation and 7AAD labeling at 48 h upon ActivinB stimulation. The percentage of cells in the different phases of the cell cycle is represented (G0/G1, S, and G2/M phases). (D and H) Percentage of apoptotic cells measured by FACS analysis of cleaved caspase-3 48 h after treatment with ActivinB. The P-values were determined by unpaired t-test and two-way ANOVA for (B and F). *P < 0.05, **P < 0.01, ***P < 0.001. Bars represent the mean ± SD. Number of replicates is n ≥ 3. The exact P-values and number of replicates are indicated in Appendix Table S5.

Source data are available online for this figure.

lines, we found heterogeneous levels of P-Smad2, from high to moderate, in the three Group 3 PDXs tested (Fig 6A). PDX4 displayed a very strong activation of the pathway, similar to that observed in the 1603MED cell line. We investigated the expression level of different mediators of the pathway by RT–qPCR (Fig 6B, Appendix Fig S4A). This analysis showed heterogeneous expression levels of *INHBB* in the 3 PDXs (Fig 6B), which tightly corresponded to the level of P-Smad2. PDX4, which showed the highest level of expression of *INHBB*, also displayed the highest P-Smad2 signal (see level of P-Smad2 in Fig 6A and INHBB expression in 6B). As in cell lines, Group 3 PDXs responded to Activin but not to TGFβ stimulation (Fig 6C). This result supported the observations in MB cell lines, suggesting a ligand specificity toward Activin in Group 3 MB. To further investigate the possibility of an autocrine mechanism involving ActivinB, we performed conditioned media experiments as described in Fig 2C. Conditioned media from PDX4, which displays a strong activation of the pathway, markedly increased P-Smad2 phosphorylation in the receiving HDMB03 cells (Fig 6D). This induction could be partially prevented when the media was pre-incubated with an ActivinB blocking antibody (Fig 6D). Moreover, PDX4 treated with the same antibody also showed a decrease in P-Smad2 (Fig 6E, Appendix Fig S4C). P-Smad2 signal could also be inhibited following treatment with inhibitors of type I receptors and follistatin (Fig 6E). We next assessed if this signaling pathway controls *PMEPA1* expression. As in cell lines, a decrease in *PMEPA1* expression was observed in PDXs after treatment with inhibitors and increased by ActivinB treatment. The expression of *MYC* and

*OTX2* remained mostly unchanged (Fig 6E, Appendix Fig S4B and C). Altogether, these results confirmed those obtained in cell lines, highlighting the presence of an autocrine stimulation involving ActivinB in Group 3 MB and identified *PMEPA1* as a gene, whose expression is controlled by this signaling pathway. We next investigated if inhibition of this pathway could be of therapeutic interest *in vivo*. The human PDX4, which displays a very high level of activation of the pathway, was orthotopically grafted into the cerebellum of nude mice. Animals were then treated 7 days per week twice a day with Galunisertib, a pharmacological inhibitor currently in clinical trial for Glioblastoma, Cisplatin as described in Niklison-Chirou *et al* (2017), or a combination. Galunisertib is described as a TGFβ type I inhibitor but, since TGFβ and Activin type I receptors are very similar, it also inhibits very efficiently ActivinB-induced Smad2 activation (Appendix Fig S1). Accordingly, we verified that Galunisertib recapitulated the main *in vitro* data obtained with LY364947 and SB431542 (Fig EV4A–C). Galunisertib-treated mice survived longer as compared to controls (Fig 7A), demonstrating the benefit of such treatment in tumors displaying high level of activation of the pathway. Accordingly, Galunisertib-treated mice displayed smaller tumors with less P-Smad2 (Fig 7B and C). No major difference was observed for Ki67 and cleaved caspase-3 staining (Fig EV4D and E). Although we did not observe any benefit from the combination of Galunisertib with Cisplatin (Figs 7A–C and EV4D and E), we cannot not exclude that different treatment kinetics could be more efficient. In this respect, other combinations with different drugs or radiotherapy remain to be evaluated.

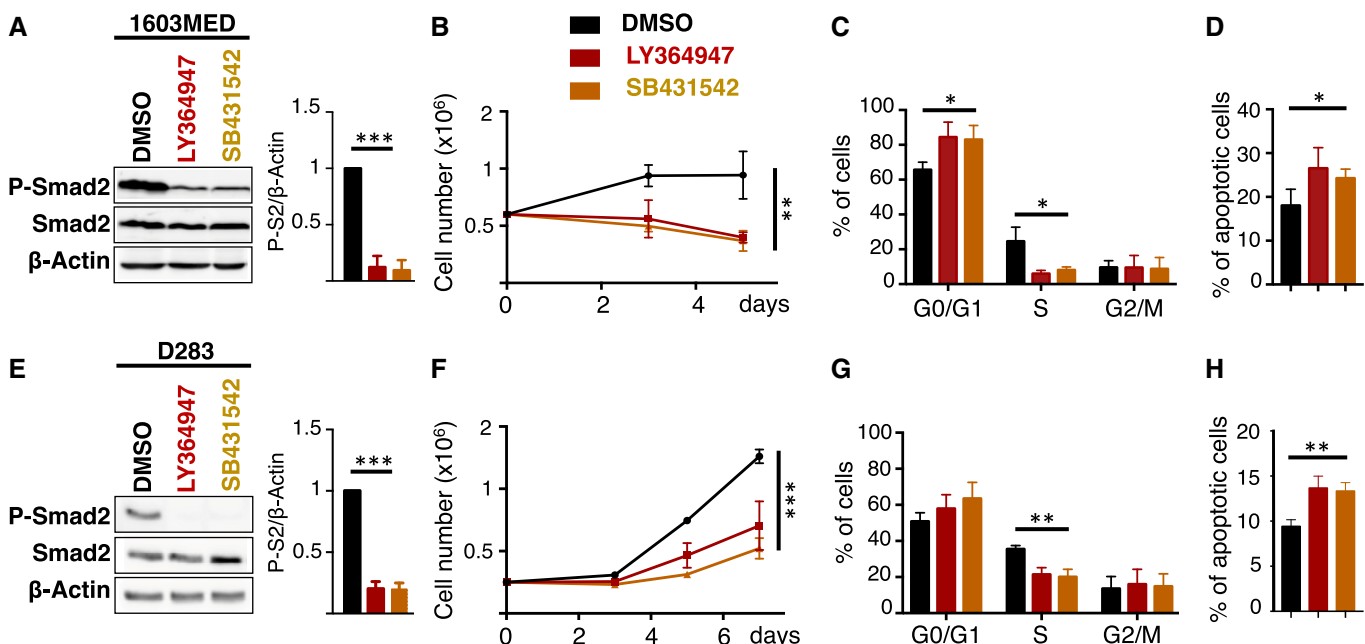

**Figure 4. TGFβ/ActivinB signaling promotes cell proliferation in Group 3 MB cell lines.**

A–H  1603MED (A-D) or D283 (E-H) cells were treated with DMSO (vehicle, black), with LY364947 (red), or with SB431542 (orange). (A and E) Immunoblot of phosphorylated Smad2 (P-Smad2), total Smad2, and β-actin upon inhibition of TGFβ/Activin signaling using LY364947 and SB431542 inhibitors for 24 h. Bar graphs on the right panel represent the quantification of the relative level of P-Smad2 (P-S2) to β-actin. (B and F) P-Smad2 to total Smad normalization is provided on Appendix Fig S5. Growth curve experiments showing cell proliferation upon TGFβ/Activin signaling inhibition. (C and G) Cell cycle analysis by FACS measuring BrdU incorporation and 7AAD labeling at 48 h upon inhibition. The percentage of cells in the different phases of the cell cycle is represented (G0/G1, S, and G2/M phases). (D and H) Percentage of apoptotic cells measured by FACS analysis of cleaved caspase-3 48 h after TGFβ/Activin signaling inhibition. The *P*-values were determined by unpaired *t*-test and two-way ANOVA for (B and F). *$P < 0.05$, **$P < 0.01$, ***$P < 0.001$. Bars represent the mean $\pm$ SD. Number of replicates is $n \geq 3$. The exact *P*-values and number of replicates are indicated in Appendix Table S5.

Source data are available online for this figure.

## TGFβ/ActivinB signaling pathway in Group 3α subtype of MB

As mentioned above, tumor samples, PDXs, and cell lines from Group 3 displayed an inter-tumoral heterogeneity regarding the level of pathway activation, some of them showing a very strong P-Smad2 basal level. Recently, intragroup heterogeneity has been described in MB (Cavalli *et al*, 2017a; Northcott *et al*, 2017; Schwalbe *et al*, 2017) with the definition of new subtypes within Group 3 tumors. We

**Figure 5. *PMEPA1* is a target gene involved in the response to Activin signaling.**

A  Ranking of top genes whose expression is correlated with *INHBB* in Group 3 MB patient samples. Spearman's rank correlation coefficient ρ and *P*-value are indicated.

B  Boxplots representing *PMEPA1* expression levels in the different MB groups (WNT in blue, SHH in red, Group 3 in yellow, and Group 4 in green) and in fetal and adult cerebellum (gray) in the dataset of Cavalli *et al* (Data ref: Cavalli *et al*, 2017b). Only *P*-values corresponding to comparisons between Group 3 and the other groups are indicated. Full statistics can be found in Appendix Table S1.

C  Scatter plot of *INHBB* and *PMEPA1* gene expression levels in all MB groups. Colored dots represent each patient samples, and colors represent the MB groups (WNT in blue, SHH in red, Group 3 in yellow, and Group 4 in green).

D  Boxplot represents the quantification of PMEPA1 protein level normalized to β-actin levels across groups.

E  Scatter plot represents log2 relative protein level of P-Smad2 (*x*-axis) and PMEPA1 (*Y*-axis) normalized to β-actin in each individual samples.

F  Immunoblots of phosphorylated Smad2 (P-Smad2), total Smad2, MYC, PMEPA1, OTX2, and β-actin were performed on extracts from 1603MED or D283 or D458 cells treated with either DMSO (vehicle), LY364947, SB431542, blocking antibody against ActivinB (Ab α-ActB), follistatin, PBS, or ActivinB (ActB) for 24 h. Blot quantification to β-actin is presented in Appendix Fig S3D.

G  Immunoblot analysis of PMEPA1 levels in 1603MED cells 48 h after transfection with either siCTRL (blue) or siPMEPA1 (red). Bar graphs on the right represent the quantification of the relative level of PMEPA1 protein level normalized to β-actin. P-Smad2 to total Smad normalization is provided on Appendix Fig S5.

H  Growth curves of 1603MED cells after transfection with either siCTRL (blue) or siPMEPA1 (red).

I  Immunoblot analysis of PMEPA1 levels in D283 cells 48 h after transfection with either siCTRL (blue) or siPMEPA1 (red). Bar graphs on the right represent the quantification of the relative level of PMEPA1 protein level normalized to β-actin. P-Smad2 to total Smad normalization is provided on Appendix Fig S5.

J  Growth curves of D283 cells after transfection with either siCTRL (blue) or siPMEPA1 (red).

Data information: The color code is the same as for (B, C, and E). Boxplot center lines show data median; box limits indicate the 25[th] and 75[th] percentiles; lower and upper whiskers extend 1.5 times the interquartile range (IQR) from the 25[th] and 75[th] percentiles, respectively (B and D). Outliers are represented by individual points (B). The *P*-values were determined by Spearman rank correlation test for (A and C), by unpaired *t*-test for (D, G and I), by two-way ANOVA for (H and J), and by Wilcoxon rank-sum test for panel (B and E). *$P < 0.05$, **$P < 0.01$, ***$P < 0.001$. Bars represent the mean $\pm$ SD. Number of replicates is $n \geq 3$. The exact *P*-values and number of replicates are indicated in Appendix Table S5.

Source data are available online for this figure.

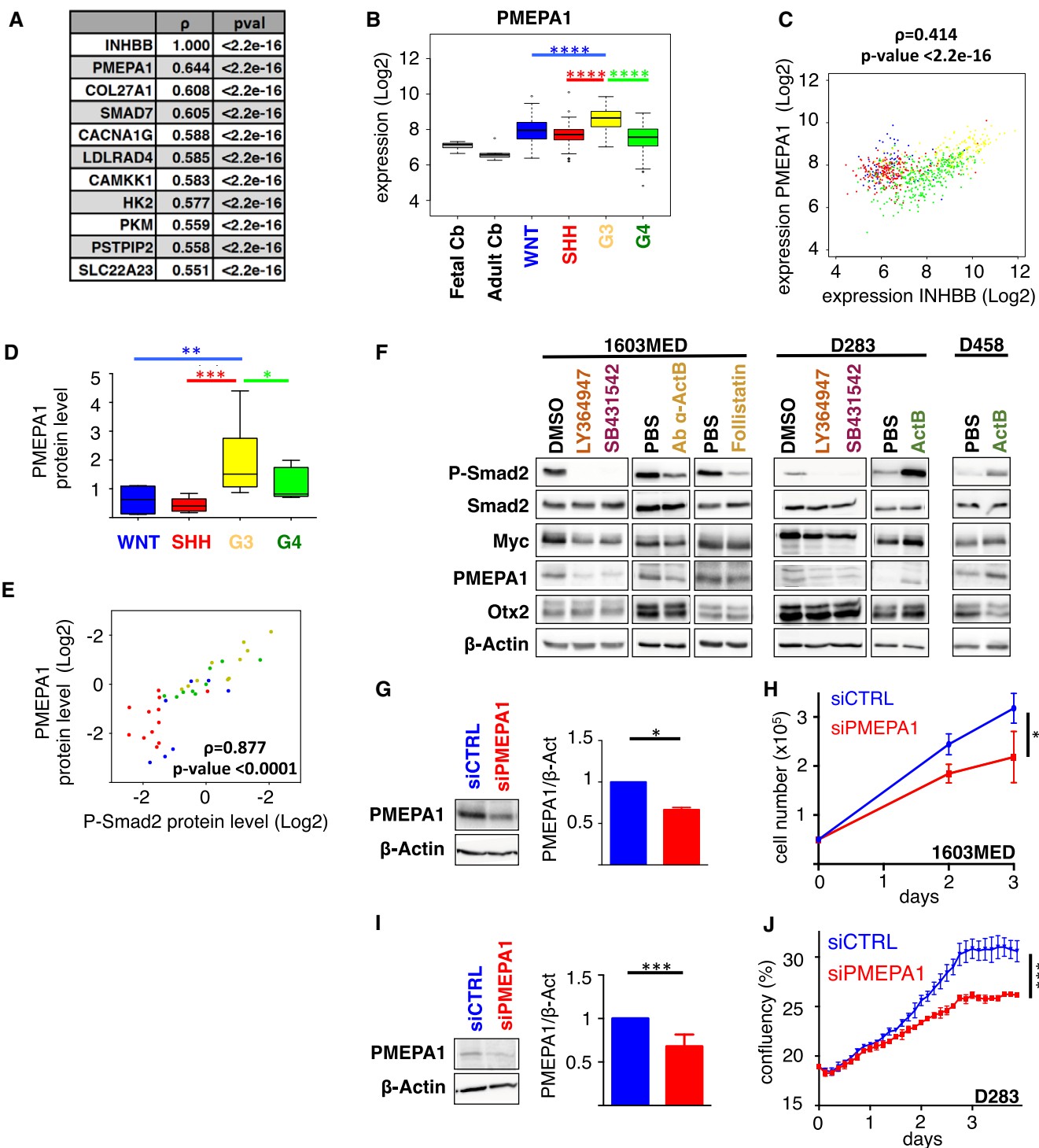

Figure 5.

wondered if this intragroup heterogeneity could explain our results. Since we showed that this strong activation was linked to an autocrine mechanism involving ActivinB, we investigated *INHBB* expression in these newly described subtypes of Group 3 tumors (Fig 7D). We found that *INHBB* displayed a significantly higher expression level in the Group 3α subtype as compared to Group 3β and Group 3γ according to Cavalli *et al* (2017a) subtyping. Interestingly, *PMEPA1* displayed the same profile, and consequently, *INHBB* and

*PMEPA1* expression was tightly correlated in Group 3 (Fig 7E). In contrast, *MYC* expression showed an opposite expression pattern as compared to *INHBB* (Fig 7D): Group 3γ subtype, which is characterized by an enrichment of *MYC* amplifications, displayed the highest *MYC* expression levels, whereas the α subtype showed the lowest (Cavalli *et al*, 2017a). We recently reported that *NRL* and *CRX* control photoreceptor genes expression and define a subset of Group 3 tumors (Garancher *et al*, 2018). We found that alike *INHBB*, *NRL* is highly

expressed in the G3α subtype (Fig 7D and Appendix Table S3). This identifies Group 3α as the subtype that expresses high level of *INHBB* and high photoreceptor genes.

## Discussion

Group 3 is the most aggressive MB group with patients showing the poorest prognosis. Several genomic alterations have been identified, including those targeting the TGFβ/Activin pathway at very low frequency. Indeed, SCNA analyses have identified uncommon gains and/or amplifications of genes encoding receptors of the TGFβ/Activin pathway. Activation of the cognate Smad2/3 pathway in Group 3 tumors has never been investigated, neither its potential biological consequences nor its potential therapeutic targeting. Using patient samples, PDXs, and cell lines, we showed that, beside these infrequent genomic alterations, the TGFβ/Activin pathway is also activated in a specific subtype of Group 3, through an autocrine

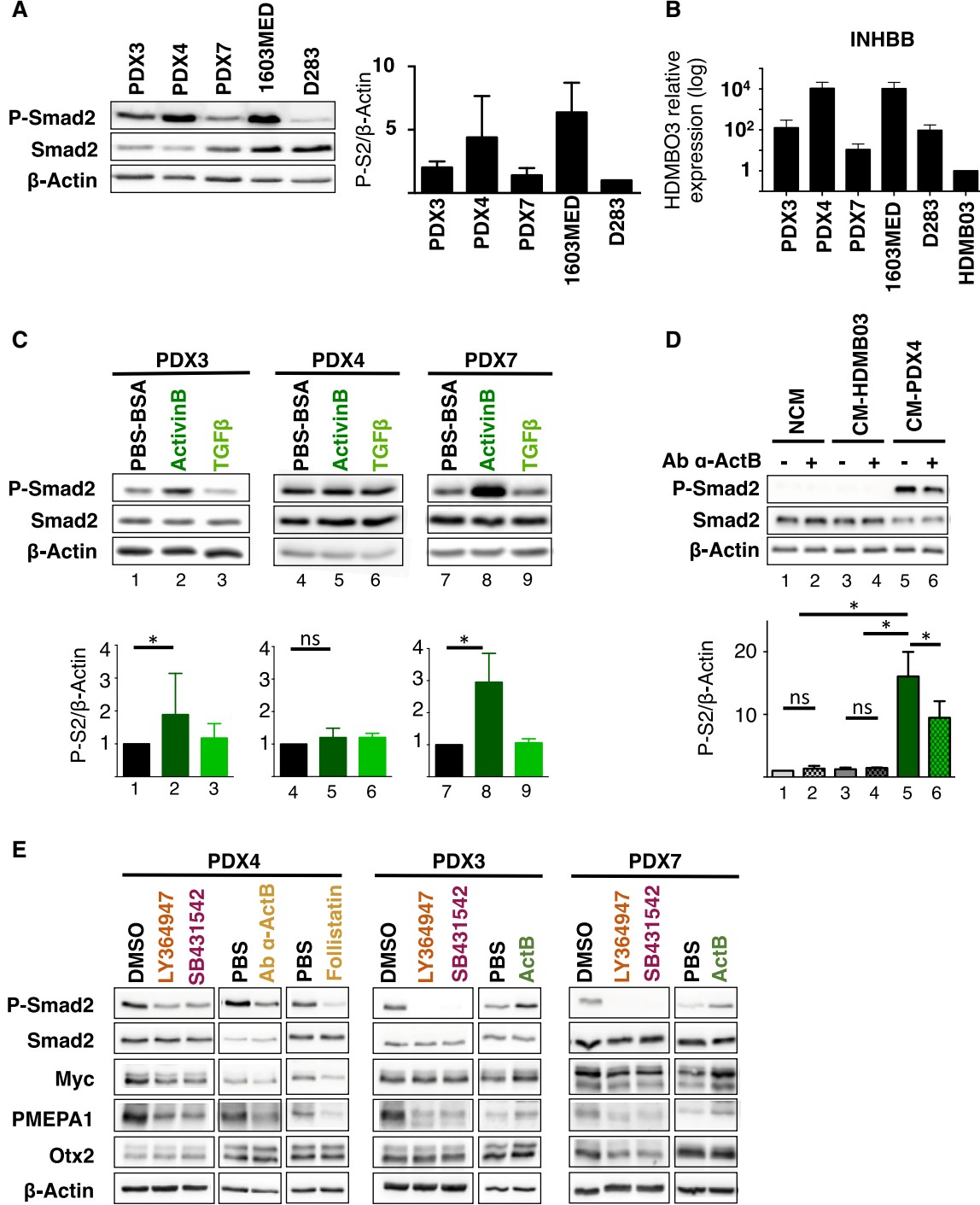

**Figure 6.**

◀

**Figure 6. Activated TGFβ/ActivinB signaling in group 3 MB-PDXs.**

A   Immunoblot analysis of phosphorylated Smad2 (P-Smad2) in Group 3 MB cell lines and PDXs. The level of total Smad2 (Smad2) was assessed, and β-actin was used as a loading control. Quantification of P-Smad2 (P-S2) to β-actin is shown on right panel.

B   Expression of *INHBB* in Group 3 MB cell lines and PDXs relative to HDMB03 (set at 1) by RT–qPCR.

C   Immunoblot of phosphorylated Smad2 (P-Smad2), total Smad2, and β-actin upon ActivinB or TGFβ stimulation for 1 h. Quantification of P-Smad2 (P-S2) to β-actin is shown below.

D   Conditioned media experiments were performed on HDMB03 MB cell line. Phosphorylation of Smad2 (P-Smad2) was analyzed by immunoblot upon treatment with either non-conditioned media (NCM), media conditioned with HDMB03 cells (CM-HDMB03), or media conditioned on PDX4 cells (CM-PDX4). Pre-incubation with blocking antibody against ActivinB (Ab α-ActB) or with vehicle was performed before HDMB03 cell-line treatment as indicated. Quantification of P-Smad2 (P-S2) to β-actin is shown below.

E   Immunoblots of phosphorylated Smad2 (P-Smad2), total Smad2, MYC, PMEPA1, OTX2, and β-actin were performed on extracts from cell cultures of PDX4, PDX3, and PDX7 treated with either DMSO (vehicle), LY364947, SB431542, blocking antibody against ActivinB (Ab α-ActB), follistatin, PBS, or ActivinB for 24 h. WB quantification is depicted in Appendix Fig S4C.

Data information: P-Smad2 to total Smad normalization is provided on Appendix Fig S5. The *P*-values were determined by unpaired *t*-test. \*$P < 0.05$. Detailed statistics are presented in Appendix Table S2 for panel (B). Bars represent the mean ± SD. Number of replicates is $n \geq 3$. The exact *P*-values and number of replicates are indicated in Appendix Table S5.

Source data are available online for this figure.

mechanism involving ActivinB. This pathway is involved in MB growth and represents an interesting therapeutic target.

### ActivinB mediates Smad2/3 signaling in Group 3 MB

While activation of the TGFβ/Activin pathway has been described in SHH group, no data are currently available regarding its activation in Group 3. A recent report showed that Prune-1 may activate the TGFβ pathway in Group 3 MB but the level of pathway activation in Group 3 was not investigated nor its functional relevance (Ferrucci *et al*, 2018). It has also been suggested that TGFβ ligands determine the promigratory potential of bFGF signaling in MB but this study was performed in non-Group 3 cell lines and in atypical MB-PDX (Santhana Kumar *et al*, 2018). Using patient samples, we showed here that the TGFβ/Activin pathway is activated in a subset of Group 3. We confirmed these results using PDXs as well as MB cell lines. In many different cancers, TGFβ pathway activation involves autocrine loops, due to the high expression of genes encoding the different TGFβ ligands (Rodón *et al*, 2014). We investigated the potential mechanism of activation of the pathway in Group 3. As in other cancers, we observed high expression of *TGFB1* and *TGFB3* in Group 3 MB. In addition, we also observed very high expression of *INHBB*, which encodes ActivinB, suggesting that TGFβ1, TGFβ3, and ActivinB ligands could be potentially responsible for pathway activation. Unexpectedly, our data clearly showed that Group 3 cells do not respond to TGFβ stimulation, while they are highly sensitive to Activin, excluding *de facto* TGFβ1 and TGFβ3 as potential ligands that would activate the pathway in an autocrine manner. The mechanism underlying the lack of TGFβ responsiveness in G3 models is currently unknown. However, we noticed a significant decrease in RNA and protein *TGFBR2* levels in G3 samples. Since TGFBR2 is absolutely required for signal transduction by TGFβ ligands, this observation may provide a plausible explanation to this lack of response. In any case, our experiments based on conditioned medium, blocking antibody, follistatin treatment, and siRNA on cell lines clearly pointed out on ActivinB as an important determinant of pathway activation in Group 3. Importantly, these observations were confirmed on PDXs. According to transcriptomic data showing that *INHBB* expression is found in a large number of Group 3 MB, this autocrine mechanism is very likely the main mechanism leading to pathway activation in this group. Additional mechanisms, such as amplifications of receptors or *Prune-1* expression (see above), could

also contribute to this activation, either by cooperating with ActivinB or by being involved in a more restricted number of Group 3 MBs that do not exhibit this autocrine mechanism. Interestingly, while TGFβs and Activins activate the same Smad pathway (Smad2/3), TGFβs autocrine mechanisms have been much more frequently described to be implicated in cancer progression than Activins (Chen *et al*, 2002; Wakefield & Hill, 2013), highlighting a singularity of Group 3 MBs. Since Activin is involved in developmental processes (Wu & Hill, 2009), its implication in Group 3 MB instead of TGFβ may relate to the pediatric nature of these tumors or to their cell of origin. In support of the latter and according to brain atlas data (http://developingmouse.brain-map.org/search/show?page_num=0&page_size=5&no_paging=false&exact_match=true&search_term=Inhbb&search_type=gene), *INHBB* displays a very cell-specific and dynamic profile during cerebellar development.

### ActivinB induces *PMEPA1* expression and promotes cell cycle progression

The TGFβ/Activin pathway is highly pleiotropic and sometimes displays antagonistic functions during carcinogenic processes. For example, it can promote either cell cycle arrest or proliferation, depending on the context. This opposite role has been well illustrated in Glioblastoma in which the epigenetic status of the cells, in particular its DNA methylation profile, is responsible for this duality (Bruna *et al*, 2007). In agreement with this pro-mitogenic activity, we found that pathway inhibition decreased cell proliferation in Group 3 MB, while ActivinB stimulation increased it by consistently promoting cell cycle progression. *MYC* and *OTX2*, two genes known to promote cell proliferation in Group 3 MB, are target genes of the Smad2/3 pathway in other contexts (Jia *et al*, 2009; Brown *et al*, 2011; Coda *et al*, 2017). In general, this signaling pathway reduces MYC expression (Warner *et al*, 1999; Seoane *et al*, 2001), although it can be induced in human embryonic stem cells (Brown *et al*, 2011). Since *OTX2* has been demonstrated to be a major Smad2/3 target gene in the nervous system (Jia *et al*, 2009), it has been proposed to be a Smad2/3 inducible gene in Group 3 MB (Ferrucci *et al*, 2018) and considered as part of this signaling pathway in MB (Northcott *et al*, 2012b). We did not detect any consistent changes in *MYC* and *OTX2* expression upon modulation of the Activin pathway, suggesting that this signaling pathway does not regulate these two genes in Group 3 tumors and promotes tumor growth through other

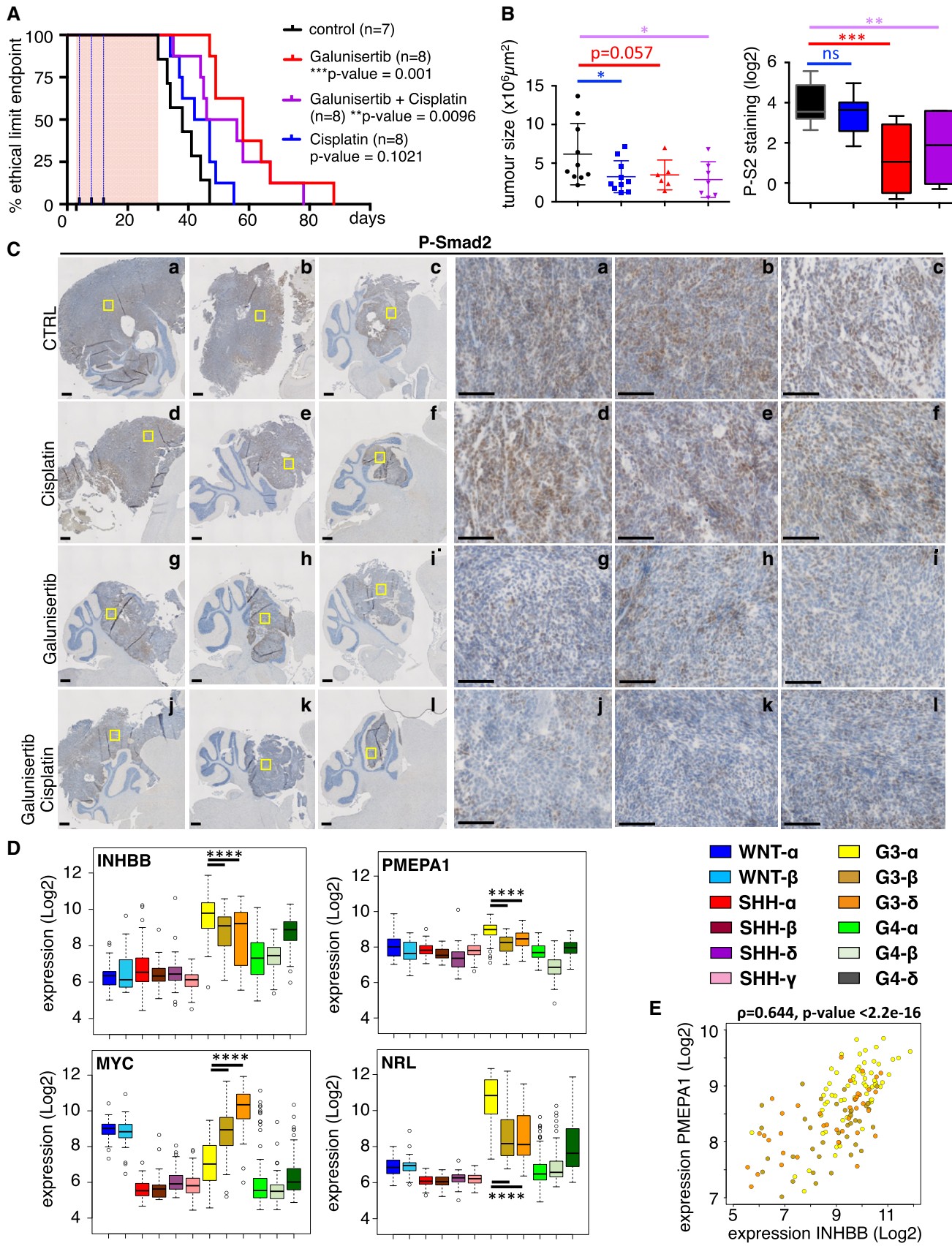

Figure 7.

**Figure 7. ActivinB signaling is a potential therapeutic target for patients of group 3 MB.**

A  Kaplan–Meier representing survival of mice treated with either vehicle (black) or Galunisertib (LY2157299, red) or Cisplatin (blue) or a combination of Galunisertib and Cisplatin (purple) after orthotopic grafting of PDX4 cells into the cerebellum. The pink rectangle represents Galunisertib treatment duration, while the blue dotted lines represent the 3 Cisplatin administrations.

B  Boxplot of tumor area after 25 days of treatment. On the right, boxplots represent quantification of P-Smad2 staining on tumor (IHC). The code color is similar to panel (A).

C  P-Smad2 staining by IHC in 3 representative tumors per group after 25 days of treatment. The scale bars represent 500 and 100 µm on the left and right panels, respectively.

D  Boxplots representing the expression level of *INHBB*, *PMEPA1*, *MYC*, and *NRL* in the different MB subtypes, as defined in Cavalli *et al* (2017a). Only *P*-values corresponding to comparisons between Group 3 subtypes are indicated. Detailed statistics are presented in Appendix Table S3. Patient samples are colored by subtypes as indicated.

E  Scatter plot of *INHBB* and *PMEPA1*, expression levels in Group 3 patient samples. Colored dots represent each patient sample, and colors represent the group 3 MB subtypes (α in yellow, β in brown, and δ in orange). Note that this panel is identical to that shown in Appendix Fig S3A (yellow) except for the color code.

Data information: Center lines show data median; box limits indicate the 25th and 75th percentiles; lower and upper whiskers extend 1.5 times the interquartile range (IQR) from the 25th and 75th percentiles, respectively (B and D). Squares represents individual tumor (B). Outliers are represented by individual points (D). The *P*-values were determined by log-rank (Mantel-Cox) test on panel (A) and unpaired *t*-test on panel (B). Wilcoxon rank-sum tests were performed for panel (D). Spearman's rank correlation coefficient ρ and *P*-value are indicated on panel E. *$P < 0.05$, **$P < 0.01$, ***$P < 0.001$, and ****$P < 0.0001$. Bars represent the mean $\pm$ SD. Number of replicates is $n \geq 3$. The exact *P*-values and number of replicates are indicated in Appendix Table S5.

mechanisms. In contrast, we showed that *PMEPA1*, whose expression is induced by TGFβ or Activin signaling in many different contexts (Coda *et al*, 2017), is also an Activin-regulated gene in Group 3 MB. Indeed, inhibition or activation of the Activin signaling pathway modulated *PMEPA1* expression accordingly. This regulation is likely to be relevant in patients since *INHBB* and *PMEPA1* expression is correlated in human MB samples. *PMEPA1* is the top correlated gene with *INHBB* within Group 3 MB, showing that their expression is strongly linked in this group. In all MB samples, the correlation is lower than within Group 3 samples. Indeed, *PMEPA1* expression is higher in Group 3 but reaches an intermediate level in WNT and SHH groups that do not express *INHBB*. In WNT and SHH groups, *PMEPA1* expression is likely due to TGFβ/Activin pathway activation, as highlighted by the high level of P-Smad2 found in patient samples in those two groups, although pathway activation is independent on ActivinB autocrine stimulation. Thus, *PMEPA1* expression likely constitutes a relevant and general readout of Smad2/3 activation, which is due to an ActivinB autocrine stimulation in Group 3 and to other mechanisms in SHH and WNT groups. The role of PMEPA1 in cancer appears to be quite complex. It has been shown to act as negative auto-regulatory loop by limiting Smad2/3 activation (Watanabe *et al*, 2010) although this appears to be isoform dependent (Fournier *et al*, 2015). Other reports suggested that PMEPA1 could promote cell proliferation in cancer cells (Vo Nguyen *et al*, 2014; Nie *et al*, 2016) and convert TGFβ/Activin signaling from a tumor suppressor to tumor promoting pathway (Singha *et al*, 2010). Although not excluding that PMEPA1 may limit Smad2/3 activation in Group 3 MB without abolishing it, our results are in line with those latter reports. Indeed, siRNA-mediated PMEPA1 downregulation decreased Group 3 cell proliferation showing that it is an important mediator of ActivinB promoting Group 3 MB growth.

## Targeting the TGFβ/Activin pathway in Group 3 as a therapeutic perspective

We observed an activation of the Smad2 pathway in Group 3 cell lines, PDXs, and patient samples. However, this activation appears to be heterogeneous. For example, some cell lines and PDXs displayed a very high basal level of Smad2 activation, while others a much more moderate and this held true on patient samples. Since different Group 3 subtypes have been described recently (Cavalli *et al*, 2017a; Northcott *et al*, 2017; Schwalbe *et al*, 2017), we investigated whether *INHBB* expression could be enriched in a given subtype. We observed that *INHBB* expression is higher in subtype Group 3α according to the classification of Cavalli *et al* (2017a). This subtype is characterized by the lack of *MYC* amplification and, as shown in this study, an overall moderate to low *MYC* expression level. This subtype displays high photoreceptor gene expression (Cavalli *et al*, 2017a), including those of the two master regulators of this program, NRL and CRX. Accordingly, we recently showed that their expression defines a specific subtype within Group 3 (Garancher *et al*, 2018). Our data may suggest that the expression of *INHBB* could lead to Smad2/3 activation in this subtype. Indeed, we found that *PMEPA1*, whose expression can be considered as a readout of Smad2/3 activation (see above), is significantly higher in Group 3α subtype as compared to other Group 3 subtypes. Moreover, its expression is tightly correlated to that of *INHBB* in the Group 3 tumors, suggesting that *INHBB* expression leads to productive pathway activation. In support to this, PDX4, which expresses very high level of *INHBB*, also displays very strong Smad2 activation. This PDX is not *MYC* amplified and highly expresses the photoreceptor genes (Garancher *et al*, 2018). It should be nevertheless mentioned that the 1603MED cell line is also characterized by high *INHBB* expression and high Smad2 activation but is *MYC* amplified and does not express high level of photoreceptor genes (Raso *et al*, 2008). Thus, we proposed that activation of the Smad2/3 pathway involving an Activin B autocrine stimulation is enriched in subtype Group 3α, although not limited to this subtype. Interestingly, treatment with Galunisertib, whose toxicity and efficacy is currently tested in clinical trials for Glioblastoma patients, increased the survival of mice orthotopically grafted with PDX4. This suggests that Group 3α patients may be particularly sensitive to pathway inhibition.

In conclusion, the TGFβ/Activin signaling is activated through an ActivinB autocrine mechanism in a subset of Group 3 MB subtype. Not only this pathway is activated, but it also plays a growth-promoting role and constitutes an important driver of therapeutic interest in these tumors. We propose that high levels of *INHBB*, PMEPA1 expression, and Smad2 phosphorylation might constitute biomarkers for potential Group 3 patients to be eligible to Galunisertib treatment.

# Materials and Methods

### Bioinformatics analyses

Normalized primary medulloblastoma gene expression data (763 samples) and samples affiliation published in Cavalli et al (2017a) were used to generate scatter plots and gene expression boxplots per subgroup and subtype for the genes of interest. Normalized primary medulloblastoma protein levels data (45 samples) and samples affiliation published in Archer et al (2018a) were used to generate protein levels boxplots per subgroup and subtype for the proteins of interest. Wilcoxon rank-sum tests were performed between subgroups and subtypes. Spearman rank correlation coefficients were computed between the INHBB gene expression values and all other genes for Group 3 samples. The gene pairs were ranked according to the Spearman correlation values.

### Patient samples

All MB samples were collected following written informed consent, and study approval was obtained by internal review boards from the following institutions: the Necker Hospital for Sick Children (Paris, France) and the Hospital for Sick Children (Toronto, Canada) (Forget et al, 2018).

### Cell culture conditions and treatments

HD-MB03 (named HDMB03) obtained from Dr. Milde (Milde et al, 2012), D458MED (named D458) obtained from Dr. Bigner (He et al, 1991), UW228 (Keles et al, 1995), ONS-76, and DAOY MB cell lines (ATCC) were cultured as described in Garancher et al (2018). 1603MED obtained from Dr. Raso (Raso et al, 2008) and D283MED (ATCC) (named D283) cell lines were maintained in DMEM condition supplemented with 12% fetal bovine serum (GIBCO), 50 units/ml penicillin and streptomycin (Invitrogen) and 0.1 mM non-essential amino acids and sodium pyruvate. 1603MED cell lines were also supplemented with 2 mM L-glutamine. All cells were cultured at 37°C in a humidified atmosphere containing 5% $CO_2$. LY363947 and SB431532 resuspended in DMSO (selleckchem) were used at a final concentration of 5 µM for 24 h. Stimulations with TGFB1 and ActivinB were performed for 1 or 24 h at 10 ng/ml. Inhibitions with a recombinant blocking antibody against ActivinB (R&D systems) or recombinant follistatin (R&D systems) were performed for 24 h at 5 and 0.2 µg/ml, respectively.

### Growth curves and proliferation assays

For growth curve analyses, 1603MED cells were plated at $8 \times 10^5$ cells/ml, and D283 and D458 at $2.5 \times 10^5$ cells/ml. Cell were treated once at day 0. Number of viable cells was assessed as indicated in each figure. For D283 and D458, proliferation was monitored using Incucyte Proliferation Assay (Essen bioscience) by analyzing the surface occupied by cells (% confluence).

### Conditioned media experiments

Receiving cells (HDMB03) were plated at $1.5 \times 10^5$ cells/well in 6-wells plates. 1603MED and HDMB03 conditioned media were obtained by 18 h of incubation at $1 \times 10^6$ cells/ml. Non-conditioned media was obtained in the same conditions in absence of cells. Media were collected, filtered, and incubated with PBS as control or blocking antibody against ActivinB (5 µg/ml) for 2 h at 4°C with rotation. Cells were treated with 1 ml of media for 1 h, and cell extracts were collected for WB analysis.

### Western Blotting and antibodies

Cell extracts were obtained and WB analyses performed as described in Rocques et al (2007). Membranes were incubated at 4°C overnight with anti-Smad2 (CST, CS86F7, 1/1,000), anti-PhosphoSmad2 (CST, CS138D4, 1/1,000), anti-OTX2 (MerckMillipore, #AB9566, 1/10,000), anti-MYC (CST, CSD3N8F, 1/1,000), anti-PMEPA1 (proteintech, 1/500), and anti-β-Actin (Sigma A1978, 1/5,000). Signals were acquired with a CCD camera (G/BOX, Syngene). All the P-Smad2/Total Smad2 normalizations for the relevant blots are provided in Appendix Fig S5.

### Real time RT–PCR

All experiments were performed according to the protocols described in Garancher et al (2018). Oligonucleotides used in this study are described in Appendix Table S4.

### siRNA and transfection assays

Transfection assays were performed in either 96- or 6-well plates. siRNA transfection was performed according to the manufacturer's instructions (Dharmacon). DharmaFECT 3 transfection reagent was used at 0.15 and 4 µl/100 µl of transfection medium for D283 and 1603MED cell lines, respectively. D283 cells were plated at $5 \times 10^5$ cells/ml and siRNA were used at a final concentration of 25 nM. 1603MED cells were plated at a concentration of $1 \times 10^6$ cells/ml with 10 µM final of siRNA. Transfection assay efficiency was assessed using siGlo (D001630-01-05). siRNA smartpool CTRL (D-001810-00-1005), smartpool INHBB (L-011702-00-0010), smartpool PMEPA1 (L-010501-00-0020), ON-TARGETplus individual siRNA PMEPA1#1 (L-010501-05), and PMEPA1#2 (L-010501-08) were purchased from Dharmacon. For rescue experiments, cells were stimulated 10 h after transfection with ActivinB at 10 ng/ml.

### Apoptosis and cell cycle analyses by flow cytometry

1603MED and D283 cell lines were plated at $8 \times 10^5$ and $2.5 \times 10^5$ cell/ml, respectively. Apoptosis was assessed at day 2 using cleaved caspase-3 staining with Apoptosis Kit, APC (BD Bioscience). Cell cycle was analyzed at day 2 using APC BrdU flow Kit (BD Bioscience). Experiments were performed using FACS Kanto (BD Bioscience) and analyzed with FlowJo software (Tree Star).

### Patient derived xenografts and PDX cultures

PDXs were obtained, maintained, dissociated, and cultured as described in Garancher et al (2018). PDX3, PDX4, and PDX7 correspond to ICN-MB-PDX-3, ICN-MB-PDX-4, and ICN-MB-PDX-7, respectively. All in vitro treatments were performed as described for cell lines.

### Animal experimentation

NMRI-nu immunodeficient mice were obtained from Janvier Laboratory. Experiments were performed on 7–8 weeks old female mice after 1 week of acclimation in animal facility of Curie Institute. Mice were housed under a controlled temperature and 12 h/12 h light–dark cycle with access to food and water *ad libitum* in conventional animal facility. For the animal welfare, mice are maintained in social groups with enrichment. Animal care and use for this study were performed in accordance with the recommendations of the European Community (2010/63/UE) for the care and use of laboratory animals. Experimental procedures were specifically approved by the ethics committee of the Institut Curie CEEA-IC #118 (Authorization 02383.02 given by National Authority) in compliance with the international guidelines.

### Orthotopic transplantation and pharmacological inhibitor treatments

NMRI Nude female mice (Janvier labs) were orthotopically grafted directly in the cerebellum at 7 weeks with $3 \times 10^5$ cells/5 µl of ICN-MB-PDX-4 cells as described in Garancher *et al* (2018). After 3 days, mice were administrated 300 µl of LY2157299 (Galunisertib, AbMole Bioscience) orally at a dose of 75 mg/kg in 12% DMSO, 30% PEG, and water. Mice were treated 7 days a week twice a day until day 30. Mice were injected with Cisplatin (Sigma) in saline solution at a dose of 2 mg/kg intraperitoneally at days 4, 8, and 12 post-grafting. Mice were euthanized when scientific and clinical end points were reached and brains were collected and fixed.

### Tissue processing and immunohistochemistry (IHC)

After 25 days of treatment, 6 mice per group received ice-cold PBS and 4% formaldehyde/PBS via intracardiac perfusions. Brains were collected and fixed overnight in 4% formaldehyde/PBS at 4°C. IHC was performed on 12-µm-thick sections with the following primary antibodies: anti-PhosphoSmad2 (CST, CS138D4, 1/300), Ki67 (CST, CS9161, 1/500), and cleaved caspase-3 (eBioscience, #14-5698-82, 1/500). Image acquisitions were performed on a Zeiss microscope. Tumor size and IHC staining were assessed using ImageJ software.

### Quantification and statistical analyses

Western blot was quantified from digital data acquisition (CCD camera) using ImageJ software. Statistical details can be found in both figures and figure legends. A $P \leq 0.05$ is considered as significant. IHC quantifications were assessed using ImageJ software. All experiments were performed, at least, in three independent triplicates. Statistical analyses are provided in Appendix Table S1 (Statistics related to Figs 1B and 5B and C, and EV1B), Appendix Table S2 (Statistics related to Figs 1D and 6B, and EV1D and Appendix Fig S4A), and Appendix Table S3 (Statistics related to Figs 7D and E, and 6B, and Appendix Fig S3A). The exact *P*-values and number of replicates for each experiment are indicated in Appendix Table S5.

**Expanded View** for this article is available online.

---

**The paper explained**

**Problem**

Medulloblastoma (MB) is a pediatric tumor of the cerebellum arising at a median age of 7 years. The current treatment associates surgery, radiotherapy, and chemotherapy and has allowed reaching an overall survival of 70–80%. MB is a heterogeneous disease classified in four groups, with the poorly characterized Group 3 showing the worst prognosis. Few recurrent genomic alterations have been identified at low frequency, and at the transcriptional level, Group 3 is known to express MYC and photoreceptor genes. While highly problematic at the clinical level, neither specific nor targeted therapy has been identified for this specific Group.

**Results**

We show that a subset of Group 3 MBs displays activation of the TGFβ/Activin pathway. In contrast to carcinomas where TGFβs are the main driver of activation of this pathway, our data established that this activation is mainly due to an autocrine stimulation involving ActivinB. We identify a subset of Group 3 tumors in which this mechanism is at play. These tumors express high levels of *INHBB* (encoding ActivinB) and display high expression of PMEPA1, a well-known target gene of this signaling pathway. Functionally, the pathway sustains cell proliferation by inducing the expression of *PMEPA1*. Importantly, treatment with Galunisertib, an inhibitor of this pathway currently tested in clinical trials for Glioblastoma patients, increases the survival of mice orthotopically grafted with Group 3 MB-PDX.

**Impact**

TGFβ/Activin signaling plays a driving role in a subset of Group 3 MBs. We propose that high level of Smad2 phosphorylation, high INHBB, and high expression of PMEPA1 could represent valuable biomarkers for identifying patients who will be particularly eligible to Galunisertib treatment.

### Acknowledgements

We thank members of our laboratory for helpful advice and comments, C. Lasgi for her assistance in FACS analyses, C. Alberti and E. Belloir at the Institut Curie mouse facilities. The authors greatly acknowledge Cédric Messaoudi from Institut Curie UMR9187/U1196 and Laetitia Besse from the PICT-IBiSA Imaging Facility for useful advices on image processing. We also thank Drs. D. Bigner, J.R. Silber, and T. Milde for providing us with materials. This work was funded by grants from Ligue Nationale Contre le Cancer (Essonne-Oise-Yvelines #M18759, #M16649, and Legs Chovet), Institut National du Cancer (INCa, Pair Pediatrie, Mr ROBOT), the IRS "NanoTheRad" of University U-PSUD (Paris-Saclay), and ADAM-Cancer (Association D'enfants Atteints de Médulloblastome). AG, LMO, and MM were supported by a fellowship from the Ministère Français de l'Enseignement Supérieur, de la Recherche et de l'Innovation, and Fondation ARC (4th year PhD fellowship).

### Author contributions

Conceptualization: CP and MM. Methodology: CP, MM, FMGC, AF, SD, AG, FB, and AE. Investigation: MM, ML, FMGC, CF, MA, AF, LM-O, SD, AG, AD, and SL. Supervision: CP, FB, and AE. Resources: CP, FD, MDT, SP, OA, FB, JM-P, OD, AR, and AE. Writing–Original draft: CP, AE, and MM. Writing–Review & Editing: CP, AE, MM, and SD. Funding acquisition: CP, FB, and AE.

### Conflict of interest

The authors declare that they have no conflict of interest.

## For more information

Website team:

(i) https://science.institut-curie.org/research/biology-chemistry-of-radiations-cell-signaling-and-cancer-axis/umr-3347-normal-and-pathological-signaling/team-eychene-pouponnot/

*In situ* Hybridization data on the mouse developing brain can be found:

(i) http://developingmouse.brain-map.org/

Public Transcriptomic analysis of MB samples (R2):

(i) https://hgserver1.amc.nl/cgi-bin/r2/main.cgi?&dscope=MB500&option=about_dscope

(ii) https://hgserver1.amc.nl/cgi-bin/r2/main.cgi

Public Proteomic analysis of MB samples:

(i) https://medullo.shinyapps.io/archer2018/

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
