## [Review Process File · EMBO Molecular Medicine]

An autocrine ActivinB mechanism drives TGF β /Activin signaling in Group 3 medulloblastoma

Morgane Morabito, Magalie Larcher, Florence M.G. Cavalli, Chloé Foray, Antoine Forget, Liliana Mirabal-Ortega, Mamy Jean De Dieu Andrianteranagna, Sabine Druillennec, Alexandra Garancher, Julien Masliah-Planchon, Sophie Leboucher, Abel Debalkew, Alessandro Raso, Olivier Delattre, Stéphanie Puget, François Doz, Michael D. Taylor, Olivier Ayrault, Franck Bourdeaut, Alain Eychène, Celio Pouponnot

Review timeline:

Submission date:	19 September 2018
Editorial Decision:	17 October 2018
Revision received:	29 April 2019
Editorial Decision:	17 May 2019
Revision received:	28 June 2019
Accepted:	02 July 2019

Editor: Lise Roth

Transaction Report:

1st Editorial Decision

17 October 2018

Thank you for the submission of your manuscript to EMBO Molecular Medicine. We have now heard back from the three referees whom we asked to evaluate your manuscript.

As you will see from the reports below, while they all mention the interest of the study, they also raise substantial concerns on your work, which should be convincingly addressed in a major revision of the present manuscript. In particular, there is a need for further strengthening of the data to fully support the conclusions and the biological relevance of the findings (adequate quantification of SMAD2 activation, robust statistics, in vitro rescue experiments, etc.).

Addressing the reviewers concerns in full will be necessary for further considering the manuscript in our journal, and acceptance of the manuscript will entail a second round of review as some claims were not fully evaluated from the current data due to limitations highlighted by the referees. EMBO Molecular Medicine encourages a single round of revision only and therefore, acceptance or rejection of the manuscript will depend on the completeness of your responses included in the next, final version of the manuscript. For this reason, and to save you from any frustrations in the end, I would strongly advise against returning an incomplete revision and would also understand your decision if you choose to rather seek rapid publication elsewhere at this stage. Should you find that the requested revisions are not feasible within the constraints outlined here and prefer, therefore, to submit your paper elsewhere, we would welcome a message to this effect.

EMBO Molecular Medicine has a "scooping protection" policy, whereby similar findings that are published by others during review or revision are not a criterion for rejection. Should you decide to submit a revised version, I do ask that you get in touch after three months if you have not completed it, to update us on the status. Please also contact us as soon as possible if similar work is published elsewhere. If other work is published, we may not be able to extend the revision period beyond three months.

I look forward to receiving your revised manuscript.

***** Reviewer's comments *****

Referee #1 (Comments on Novelty/Model System for Author):

Adequate mouse models have been used in the study.

Referee #1 (Remarks for Author):

In this study, the authors report that activin signaling is activated in some tumor samples and cell lines of Group 3 medulloblastoma. This activation may be caused by an autocrine mechanism as these tumors and cells produce more activinB. They further show that activin promotes cell proliferation, which may be mediated by PMEPA1, a known TGF β target. In the last, they show that the PDX with high activin signaling activity can respond to Galunisertib, a pharmacological inhibitor currently in clinical trial for glioblastoma. Overall it is an interesting study with solid data. However, some important questions remain unclear. Specific concerns are as follows:

1. Fig 1: More tumor samples are needed to support the conclusion. The statistic methods should be indicated (Also for other figures).
2. Fig 1A: Need to show total Smad2 expression in these tumor samples. The method to calculate the relative p-Smad2 levels has to be clearly described in the figure legend. Smad2 activation levels should be measured by the relative levels of p-Smad2 versus total Smad2 after normalized with the loading control actin.
3. Fig 4 and 5: The data showed that modulation of activin signaling also affected cell apoptosis, suggesting that activin signaling is important for cell survival. What is the underlying mechanism?
4. It seems that MB tumors and cells express TGF β receptors at the similar levels as activin receptors (Fig 1B), why they don't respond to TGF β in Smad2 activation (measured by p-Smad2) in Fig 3A and 6D?
5. Some important information is missing in the text. For instance, the authors should clearly indicate that Galunisertib is a TGF β type I receptor inhibitor.
6. In the title, the "photoreceptor subtype" is used. In fact, this study did not show any link between activin signaling with photoreceptor expression. Therefore, the current title is confusing, "photoreceptor" should not be emphasized in the title.

Referee #2 (Remarks for Author):

This is an interesting story with good potential to make an impact in the field. However, results are preliminary and incomplete and the manuscript seems to have been finalised in a rush and attention to details is lacking. This is unfortunate as it gives the impression the experiments are carried out at a low bar; this is potentially not the case but authors have to convince reviewers that their data are indeed robust and significant.

Major points to be addressed:

Figure 1

- 1A: It is essential the expression level of total SMAD2 are shown (as they do for the following p-SMAD2 blots). Once the total level of SMAD2 are evaluated, they need to quantify again the p-SMAD2 signal normalizing it to the total amount of SMAD2 and not only to the beta-actin. At present the claim that there is an increased phosphorylation in GR3 is not supported by convincing evidence.

Also, the sentence "this established that TGF β /Activin pathway is also activated in Group 3 and WNT MB patients" is not justified taking into account the data presented. From the boxplot it is not clear whether the pathway is activated in WNT group. Moreover because the p-value is not shown, it is unclear whether any purported difference is significant or not.

- Figure 1B: p-values are absent in these plots and a normal control is lacking. So, although the expression of TGFB1, TGFB3 and INHHB seems to be increased in GR3, it is difficult to

understand why the authors focused their analysis on these genes. Is the increased expression shown statistically significant? How many times were the experiments performed?

- The authors state that amplifications of the receptors are present in less than 10% of GR3 patients, are these the same samples analysed previously?

Figure 2 and Figure S1

- Fig 2A: The basal levels of p-SMAD2 in the cell lines used by the authors are highly variable between the different experiments. See for example: (i) a discrete p-SMAD2 band is seen in fig.2A for D458 but a complete lack of SMAD2 phosphorylation is shown in fig.3A, (iii) an increased phosphorylation compared to the other GR3 cell lines is shown for 1603MED in fig.2A while in fig. 3A the level of p-SMAD2 is similar compared to the other cells and (iii) D286 shows phosphorylation of SMAD2 in fig.2A while in fig. 4E there is no p-SMAD2 signal in the control sample (PBS). It is unclear how these different results can be reconciled. What is happening to the cells throughout the different experiments? And what is the cause of this variability? Also, it is absolutely necessary that authors clarify how many time the experiments have been repeated. The number of repetitions (n) is not mentioned for any of the experiments presented and there is no quantification either.

- Fig 2B to F and Fig S1: Authors need to clarify the number of biological/technical replica they did and add p-values, at least for the genes where a claim of increased/decreased expression (e.g. ACVR1B, ACVR2A, ACVR2B, TGFBR2, TGFB3 and INHBB) is made.

- INHBB shows quite high expression (although p-value and a normal control are lacking) also in GR4 samples, it might be interesting to check SMAD2 phosphorylation also in GR4 cell lines.
- The authors should check the expression of the genes that they state as involved in the regulation of the signalling pathway (e.g. ACVR1B, ACVR2A, ACVR2B, TGFBR2, TGFB3 and INHBB) at protein level.

Figure 3

- Again the number of biological replica for all the experiments presented here is missing and the quantification needs to take into account the total level of SMAD2.

- Fig 3D-E-F: The author should stimulate with ActivinB the siRNA-treated cells to check for a rescue in the phenotypes presented and exclude any off target effect of the siRNA.

- The authors discuss that Prune-1, recently described as involved in TGF-beta activation in GR3 MB, can contribute/cooperate in the pathway activation that they have described. The effect of ActivinB on Prune-1 expression level should be assessed.

Figure 4

- Again no mention of the number of biological replica carried out and non quantification of the results with p-values. D458 and D283 show a complete absence of p-SMAD2 in the basal level represented by PBS-treated sample, while in Fig. 2A they seem to express a completely different level of SMAD2 phosphorylation.

Figure 5

- As above, number of biological replica, quantification and p-values missing.

Figure 6

- As above, number of biological replica, quantification and p-values missing.

- Fig 6C: Authors should show the correlation coefficients for each different subgroups to be able to understand if PMEPA1/INHBB correlation is really important just for GR3 and to justify their sentence "PMEPA1 is the top correlated gene with INHBB within Group 3 MB, showing that their expression is strongly linked in this group. In all MB samples, the correlation is lower than within Group 3 samples".

- Fig 6H to 6K: Authors should reconstitute PMEPA1 expression in siRNA-treated cells to show a rescue of the phenotypes presented and exclude off target effects of the siRNA.

- The authors suggest that "PMEPA1 is an important mediator of Activin signaling-mediated proliferation in Group 3 MB". PMEPA1 protein expression level needs to be checked after ActivinB stimulation and inhibition (blocking antibody) for the claim to be justified.

- It would be extremely interesting to see the level of PMEPA1 in patients analysed in Fig.1A to strengthen the correlation proposed by the authors.

Figure 7 and Figure S2

- As above, number of biological replica, quantification and p-values missing.

- Fig.7A: The authors report that they "found significant but heterogeneous levels of P-Smad2 in the three Group 3 PDX tested". Is not clear how they can define these levels as significant since there isn't any quantification or p-value. Moreover they should show non-GR3 PDX as a control to show whether there is a GR3-specific increased phosphorylation of SMAD2 also in the PDX model.

- Fig. 7G: Protein levels of PMEPA1, OTX2 and MYC after the treatments must be showed.

- Fig. 7H: The authors should show pSMAD2, PMEPA1, ACTIVIN B expression in tumour after the treatment with Galunisertib, which is not clear if it's a specific inhibitor of Activin B.
- Fig. 7I and 7H: The authors should check the GR3 subtype of the PDXs that they used. This would strength the correlation that they are proposing with the alpha subtype.
- The authors should describe what is presented in figure S2 and show p-values for it.
- To state that "This identifies the α subtype as the photoreceptor Group 3, which displays activation of the Smad2 pathway by an ActivinB autocrine mechanism" the authors need to: 1) check correlation of NRL-CRX-INHBB in all the subgroups and subtypes. Are they correlated just in GR3 alpha?
2) Check the subtype affiliations of the PDX they used and have PDX from different subgroups and subtype to demonstrate that what they describe holds true only in the GR3 alpha PDXs.
3) Check in recently published proteomic data whether there is an increased phosphorylation of p-SMAD2 in GR3 tumours and if this is specific of the alpha subtype.

Referee #3 (Comments on Novelty/Model System for Author):

The data is novel as the TGF-beta pathway has not been explored mechanistically in medulloblastoma molecular subgroups before. The medical impact for individualized subgroup-specific therapy is probably low as the Activin pathway is in need of more specific inhibitors.

Referee #3 (Remarks for Author):

Morabito et al. are presenting a manuscript in where the TGF-beta/Activin pathway is explored in the most aggressive subgroup of Group 3 medulloblastoma. They suggest that it is rather Activins than TGF-beta proteins that are potential targets for a subtype of Group 3-alpha patients. They further present how autocrine stimulation of secreted Activin B can stimulate cells via PMEPA1 and that downstream Smad2 pathway activity can be suppressed by Activin B blocking antibodies or pharmacological pathways inhibitors in vitro and also in vivo. The data is interesting but as some critical experiments and relevant inhibitors for the Activin pathway are lacking the results do not provide compelling evidence that the Acitivin pathway is especially warranted for MB subtype 3-alpha.

Main concerns:

1. In Fig1B it is not clear if differences are statistically significant. It is sufficient to highlight the ones that are significant in main Figure and put the rest in a Supplement Figure.
2. Regarding Figure 2B-F + Figure 7B. All the RT-PCR data have been normalized to HDMB03 (probably because it has lowest P-Smad2 of the Gr3 cell lines) - however this is stated nowhere in the figure legends, methods, etc. Further, X-axis should have the same range for all graphs for better comparisons.
3. Regarding Figure 2C. Do not agree with the author's statement that INHBB is generally higher expressed in Gr3 cell lines than non-Gr3 cell lines (only true for lines 1603MED and D283).
4. In Figure 3C, HDMB03 responds to conditioned media from 1603MED by phosphorylating Smad2; but the authors don't follow up on whether TGF-beta signaling activation has functional consequences on the treated cells (e.g. increase in proliferation) and whether this can be prevented by the anti-Activin B antibody.
5. Regarding Figure 3F & 6I. It is impossible to show proliferation with basically one time point (apart from start point); authors could have at least added one additional time point (Day 1); Guess this is also a transient effect or is cell proliferation inhibited after Day 3?
6. In Figure 4, the data for HDMB03 is missing (lowest basal activation of the Activin pathway) - should benefit the most from exogenous Activin B. Could also not find any information how often cells were treated with Acitivin B. Please clarify this.

7. In Figure 6, it is confusing why authors are focus on PMEPA1 at all. What role does it play for the story of the paper (there's no further focus on PMEPA1 in the remaining paper)? Is this protein druggable or downstream of pSmad2? Suggested experiment: treat HDMB03 with Activin B and show upregulation of PMEPA1 and further check for pSmad2.

7. A major problem is that the drugs used in the manuscript are all TGFRB1 inhibitors and not specific ACVR inhibitors. Both pathways will stimulate downstream pSmad2 as they show with either TGF-beta or Activin B stimulation. Would thus be important to find an ACVR specific drug (ALK4/7 specific) or interesting to see how e.g. Decorin or FST treatment would affect tumor growth, or use TGF-beta stimulation in combination with available drugs that target ALK4, 7 as well as ALK5. The blocking antibody itself would be interesting to test in vivo or valid reasons for not using this should be explained.

8. The TGFBR1 inhibitor Galunisertib is poorly introduced and it is not clear why authors don't show data on this in vitro as they do with LY364947 and SB431542? Does Galunisertib have the same effects as LY364947 and SB431542 (e.g. on proliferation, P-Smad2 levels etc.). In the figure it should also be clearly shown that the treatment was stopped during weekends (no continuous treatment).

9. In Figure 7H, it is strange that there's no in vitro data for PDX7 (used for in vivo experiment), only for PDX3 and PDX4. Would mice benefit from long-term treatment? How does the Galunisertib treatment perform in comparison to standard treatment options (e.g. Cisplatin)? What are the actual effects on the tumour? Smaller? More/less apoptosis? Less proliferation/cell cycle arrest/differentiation? What about combination treatments? From the in vitro data I assume tumor growth is slowed down, but tumour cells are not killed (a second drug could perhaps do the job).

10. In Figure 7I-J, I don't think authors can proof that Acitivin pathway inhibition is especially warranted for subtype 3alpha although the correlations to NRL and CRX exist. What if you treat a SHH PDX with Galunisertib? Is the drug really specific to Group 3? They only have in vivo results from one Gr. 3 PDX and the cell lines do not respond to Activin/TGF signalling perturbations corresponding following their MYC, p53 or INHBB status. Authors don't know why some Gr3 have higher basal TGFbeta activity than others.

Minor concerns:

1. There are a few spelling errors/grammar mistakes:

End of abstract text: orthotopically

Introduction: photoreceptor genes in which few amplifications...

A few sentences in the first section of page 4 are oddly written. The description of the subtypes on page 5 should be clearer and better written with regard to the 3 distinct Gr3 subtypes in the Cavalli set.

Page 9: ...decreased of cell growth (Figure 3F).

Material/method: Dr. Milde is misspelled

Figure 7 legend: Kaplan-Meier misspelled

2. Put Figure 7G in a Supplementary figure.

3. In Figure 4F: Cells stop proliferating at day 4 (plateau); interesting to see what happens if you keep adding Activin.

4. In Figure 6K. Why do control D283 cells stop proliferating after 80 hrs? Cannot be lack of space, because only 30% confluent according to Incucyte.

1st Revision - authors' response

29 April 2019

Referee #1 :

In this study, the authors report that activin signaling is activated in some tumor samples and cell lines of Group 3 medulloblastoma. This activation may be caused by an autocrine mechanism as

these tumors and cells produce more activinB. They further show that activin promotes cell proliferation, which may be mediated by PMEPA1, a known TGF β target. In the last, they show that the PDX with high activin signaling activity can respond to Galunisertib, a pharmacological inhibitor currently in clinical trial for glioblastoma. Overall it is an interesting study with solid data. However, some important questions remain unclear. Specific concerns are as follows:

1. Fig 1: More tumor samples are needed to support the conclusion. The statistic methods should be indicated (Also for other figures).

As requested, we have added more samples from each group for a total of 38 tumors. We have now 7 WNT tumors, 12 SHH, 10 G3 and 9 G4 (see Figure 1A and below). The statistical methods are included in the legends as well as for all the other figures.

Figure 1A

2. Fig 1A: Need to show total Smad2 expression in these tumor samples. The method to calculate the relative p-Smad2 levels has to be clearly described in the figure legend. Smad2 activation levels should be measured by the relative levels of p-Smad2 versus total Smad2 after normalized with the loading control actin.

We have added total Smad2 as requested. In addition to previous P-Smad2/actin normalization, we have added P-Smad2/total-Smad as requested by the reviewer (Figure EV1A). However, regarding quantification of Smad2 activation, we strongly believe that P-Smad2/Actin normalization is also relevant since it gives crucial information regarding the overall number of Smad2 molecules that are active, enter the nucleus and are available for target gene activation as compared to the normalization to total Smad2. The P-Smad2/totalSmad2 provides information regarding the ratio between active vs non active Smad2 (ie nuclear vs cytoplasmic) but is not representative of the ability of Smad2 to induce a biological response. For example, if all Smad2 molecules are phosphorylated in cells expressing very low amount of total Smad2, the number of active nuclear Smad2 molecules would be very low as well, and therefore not sufficient to significantly activate target genes.

To illustrate our statement, one can focus on tumor samples #34 and #351 from the WNT group. The choice of quantification will drastically change the interpretation. If normalized to total Smad2, samples #351 should be considered to show the most active pathway although P-Smad2 is barely detectable (relative level 4.98, see figure in point 1 or normalization in Figure 1A). In contrast, sample #34, which displays a robust P-Smad2 signal would be considered with low pathway activation (relative level 1.09). We do not believe correct to conclude that the pathway is more than 4.5 fold active in sample #351 than in sample #34.

In contrast to other signaling pathways where the overall level of effectors expression is usually unchanged, the Smad2 protein level has been shown to be regulated upon pathway activation in particular by an autoregulatory mechanism explaining the observed variations.

For all these reasons, we strongly believe that normalization of P-Smad2 to β -actin is highly representative of pathway activation, as discussed in the result section (see also response to point 18 of referee#2). To provide as much information as possible to the reader, both quantifications were included (below the WB (see Figure 1A (or figure shown in point #1 and Figure EV1A (and below).

Figure EV1A

3. Fig 4 and 5: The data showed that modulation of activin signaling also affected cell apoptosis, suggesting that activin signaling is important for cell survival. What is the underlying mechanism? It should be noticed that protection from apoptosis is not observed in all cell lines. For example, no protection is observed in D283 upon activin stimulation (Figure 3H). Moreover, in cell lines in which activin protects from apoptosis, the effect is very modest (below 1.5x in every condition with, in general, less than 10%-15% of apoptotic cells). For these reasons, it is difficult to chase the underlying mechanism, and this would not be much informative since apoptosis does not play a major role in the described effects.

4. It seems that MB tumors and cells express TGF β receptors at the similar levels as activin receptors (Fig 1B), why they don't respond to TGF β in Smad2 activation (measured by p-Smad2) in Fig 3A and 6D?

The data represent relative but not absolute levels. In this context, it is not possible to comparatively assess which receptors are more expressed. Nevertheless, some observations may explain this lack of response. Both TGFBR1 and TGFBR2 are necessary for pathway stimulation in response to TGF β ligands. We observed that G3 tumors expressed lower levels of *TGFBR2* both at the mRNA level (below and Figure EV1B) and at the protein level in the Archer dataset (Archer et al. Cancer Cell, 2018) (see below and Figure EV1C). The same is observed in cell lines and PDXs (see Figure EV1D and Appendix Figure S3A or below). Although likely not the unique mechanism, the low level of TGFBR2 may explain the lack of TGF β response. This has now been included in the main text of the manuscript.

Figure EV1B and EV1C

RNA (left) and protein (right) level of TGFBR2 across groups

RT-qPCR assessing the relative level of TGFBR1 and TGFBR2 in cell lines (blue non Group 3 cell lines, yellow Group 3 cell lines and PDXs) (please note the log scale). This figure has not been included in the manuscript but is provided to the reviewer to illustrate our answer. If the reviewer believes that this figure should be included, we are willing to do it.

5. Some important information is missing in the text. For instance, the authors should clearly indicate that Galunisertib is a TGF β type I receptor inhibitor.

This has now been clearly stated and the text has been modified accordingly. To clearly show that Galunisertib inhibits TGF β type I receptor as well as Activin type I receptor, we added a new supplementary figure (Appendix Figure S1 and below) showing that Galunisertib, alike LY364947 and SB431542, inhibits both TGF β - and Activin-induced P-Smad2. We also demonstrated the specificity of the anti-activinB blocking antibody by showing that it inhibits P-Smad2 induced by ActivinB but not by TGF β .

Appendix Figure S1

6. In the title, the "photoreceptor subtype" is used. In fact, this study did not show any link between activin signaling with photoreceptor expression. Therefore, the current title is confusing, "photoreceptor" should not be emphasized in the title.

As requested by the reviewer, the title was modified. The new title " An autocrine ActivinB mechanism drives TGF β /Activin signaling in Group 3 medulloblastoma" does not refer to photoreceptor anymore. We have also modified the abstract accordingly.

Referee #2 (Remarks for Author):

This is an interesting story with good potential to make an impact in the field. However, results are preliminary and incomplete and the manuscript seems to have been finalised in a rush and attention to details is lacking. This is unfortunate as it gives the impression the experiments are carried out at a low bar; this is potentially not the case but authors have to convince reviewers that their data are indeed robust and significant.

Major points to be addressed:

Figure 1

1/ • 1A: It is essential the expression level of total SMAD2 are shown (as they do for the following p-SMAD2 blots). Once the total level of SMAD2 are evaluated, they need to quantify again the p-SMAD2 signal normalizing it to the total amount of SMAD2 and not only to the beta-actin. At present the claim that there is an increased phosphorylation in GR3 is not supported by convincing evidence.

See response to reviewer#1 point#2 (paste below)

We have added total Smad2 as requested. In addition to previous P-Smad2/actin normalization, we have added P-Smad2/total-Smad as requested by the reviewer (Figure EV1A). However, regarding quantification of Smad2 activation, we strongly believe that P-Smad2/Actin normalization is also relevant since it gives crucial information regarding the overall number of Smad2 molecules that are active, enter the nucleus and are available for target gene activation as compared to the normalization to total Smad2. The P-Smad2/totalSmad2 provides information regarding the ratio between active vs non active Smad2 (ie nuclear vs cytoplasmic) but is not representative of the ability of Smad2 to induce a biological response. For example, if all Smad2 molecules are phosphorylated in cells expressing very low amount of total Smad2, the number of active nuclear Smad2 molecules would be very low as well, and therefore not sufficient to significantly activate target genes.

To illustrate our statement, one can focus on tumor samples #34 and #351 from the WNT group. The choice of quantification will drastically change the interpretation. If normalized to total Smad2, samples #351 should be considered to show the most active pathway although P-Smad2 is barely detectable (relative level 4.98, see figure in point 1 or normalization in Figure 1A). In contrast, sample #34, which displays a robust P-Smad2 signal would be considered with low pathway activation (relative level 1.09). We do not believe correct to conclude that the pathway is more than 4.5 fold active in sample #351 than in sample #34.

In contrast to other signaling pathways where the overall level of effectors expression is usually unchanged, the Smad2 protein level has been shown to be regulated upon pathway activation in particular by an autoregulatory mechanism explaining the observed variations.

For all these reasons, we strongly believe that normalization of P-Smad2 to β -actin is highly representative of pathway activation, as discussed in the result section (see also response to point 18 of this referee). To provide as much information as possible to the reader, both quantifications were included (below the WB (see Figure 1A (or figure shown in point#1 and Figure EV1A (and below).

Figure EV1A

2/ Also, the sentence "this established that TGF β /Activin pathway is also activated in Group 3 and WNT MB patients" is not justified taking into account the data presented. From the boxplot it is not clear whether the pathway is activated in WNT group. Moreover because the p-value is not shown, it is unclear whether any purported difference is significant or not.

This has now been deleted from the text. We only mentioned that some Gr3 tumors display high level of pathway activation: "...led us to conclude that TGF β /Activin pathway is activated in some Group 3 patients."

Full statistics are now included (Figure EV1A and Appendix Table S1) as well as statistical methods in the legends. These are now included in all the other figures.

3/ • Figure 1B: p-values are absent in these plots and a normal control is lacking. So, although the expression of *TGFB1*, *TGFB3* and *INHBB* seems to be increased in GR3, it is difficult to understand why the authors focused their analysis on these genes. Is the increased expression shown statistically significant? How many times were the experiments performed?

Previous Figure 1B (New Figure 1B (3 boxplots) and Figure EV1B) represents data obtained from transcriptomic datasets described in Cavalli et al. that include more than 600 samples. These samples are different from those presented in Figure 1A. The statistical power of this analysis is based on the number of samples analyzed. Accordingly, the same results are found in the Pfister dataset (<https://hgserver1.amc.nl/cgi-bin/r2/main.cgi>).

As mentioned in the text ("Activation of the *Smad2/3* pathway in cancer is frequently due to autocrine/paracrine activation by *TGF β* ligands (Rodón et al., 2014). Therefore, we analyzed the expression of major mediators of the *TGF β /Activin* pathway, including ligands and receptors in previously published MB dataset at the mRNA (Cavalli et al. 2017) and protein (Archer et al., 2018) levels. "), we analyzed the expression of the different ligands in this dataset to identify mediators of the pathway that could be overexpressed in G3 and explain the high level of P-Smad2. We found that *INHBB*, *TGFB1* and *TGFB3* are highly expressed in G3 and could be, therefore, responsible for this activation. As requested, statistical tests were performed and show that these increases are statistically significant. The most important statistical informations are embedded in the Figures (see Figure 1B and Figure EV1) while the full analyses are included as a supplemental table (Supplemental Table S1).

4/ • The authors state that amplifications of the receptors are present in less than 10% of GR3 patients, are these the same samples analysed previously?

In fact, very rare amplifications of the receptors have been described from a huge number of samples. Interestingly, while few *ACRV2A* amplifications are present in the Cavalli dataset, they are found in G3 α (see below data extracted from this article Cavalli et al., Cancer Cell 2017).

Data extracted from Cavalli et al., 2017 « Significant regions of focal SCNA identified by *GISTIC2* in a subtype-specific analysis for Group 3 samples. Only peaks with residual q value < 0.05 are annotated. Genes affected by focal copy number that are mentioned in the main text are colored ». Cavalli et al. 2017.

Figure 2 and Figure S1

5/ • Fig 2A: The basal levels of p-SMAD2 in the cell lines used by the authors are highly variable between the different experiments. See for example: (i) a discrete p-SMAD2 band is seen in fig.2A for D458 but a complete lack of SMAD2 phosphorylation is shown in fig.3A, (iii) an increased phosphorylation compared to the other GR3 cell lines is shown for 1603MED in fig.2A while in fig.

3A the level of p-SMAD2 is similar compared to the other cells and (iii) D286 shows phosphorylation of SMAD2 in fig. 2A while in fig. 4E there is no p-SMAD2 signal in the control sample (PBS). It is unclear how these different results can be reconciled. What is happening to the cells throughout the different experiments? And what is the cause of this variability? Also, it is absolutely necessary that authors clarify how many time the experiments have been repeated. The number of repetitions (n) is not mentioned for any of the experiments presented and there is no quantification either.

Previous Figures 2A, 3A, 4E now correspond to Figures 1C, 2A, 3E respectively (as requested by reviewer#3). The variability of basal level of P-Smad2 in these different panels is due to different time of WB exposure and should not be compared. We chose a shorter exposure in the former Figures 2A and 3E to better highlight the P-Smad2 stimulation by avoiding signal saturation. To fulfill the reviewer's request, we now include longer exposures (new Figure 2A and 3E) similar to those presented in the novel Figure 1C.

All the experiments presented in the manuscript have been performed at least 3 times and as requested quantification is now presented for each blot. Number of replicates is now stated in each Figure legend.

6/ • Fig 2B to F and Fig S1: Authors need to clarify the number of biological/technical replica they did and add p-values, at least for the genes where a claim of increased/decreased expression (e.g. ACVR1B, ACVR2A, ACVR2B, TGFBR2, TGFB3 and INHBB) is made.

All the experiments presented in the manuscript have been performed at least 3 times and as requested p-values were added in the different figures or in Appendix Table S1 to S3. Number of replicates is stated in the Figure legend.

7/ • INHBB shows quite high expression (although p-value and a normal control are lacking) also in GR4 samples, it might be interesting to check SMAD2 phosphorylation also in GR4 cell lines. p-values are now added and data are normalized to HDMB03. We agree that it would have been interesting to check Smad2 phosphorylation in G4 cells, but while a number of G3 cell lines have been described, no *bona fide* G4 cell lines have been reported so far.

8/ • The authors should check the expression of the genes that they state as involved in the regulation of the signalling pathway (e.g. ACVR1B, ACVR2A, ACVR2B, TGFBR2, TGFB3 and INHBB) at protein level.

Unfortunately, no suitable antibody is available for these proteins. We tested several antibodies but none of them gave a robust and clear signal. Below is shown one of our WB attempts with TGFBR1 (~56kDa) and TGFBR2 (predicted 65kDa but migrates above 70kDa) antibodies in the indicated cell lines (the 72kDa and 55kDa bands are indicated on the MW ladder). As observed, we were unable to unambiguously identify the band corresponding to these receptors. This was also true for the other receptors.

Western Blots probed with TGFBR1 or TGFBR2 antibodies

Nevertheless, to meet the reviewer's concern, we used public proteomic dataset (Archer et al., 2018) to analyze the level of different receptors across groups. We confirmed the data observed at the

protein level and, importantly, we observed an increase of *INHBB* in G3. These data are now added in the revised version as Figure EV1C.

Figure EV1C

Figure 3

9/ • Again the number of biological replica for all the experiments presented here is missing and the quantification needs to take into account the total level of SMAD2.

This is now included in the revised version of the Figure (new Figure 3) and the number of replicates are indicated in the Figure legend.

10/ • Fig 3D-E-F: The author should stimulate with ActivinB the siRNA-treated cells to check for a rescue in the phenotypes presented and exclude any off target effect of the siRNA.

The requested rescue experiment has been performed and is now included as a new panel in the revised version of the manuscript (Figure EV2 and see below). ActivinB treatment rescued the P-Smad2 level, PMEPA1 induction and the defect in cell proliferation (Figures EV2 and EV3A). The text has been modified accordingly. P-values are added and the number of replicates is stated in the Figure legend.

Figure EV2

Figure EV3A

11/ • The authors discuss that Prune-1, recently described as involved in TGF-beta activation in GR3 MB, can contribute/cooperate in the pathway activation that they have described. The effect of ActivinB on Prune-1 expression level should be assessed.

We did not observe any change in the expression of *Prune-1* by RT-qPCR upon inhibition or stimulation with ActivinB in cell lines and PDXs (see below). Since these results are negative and in

order to keep the manuscript as simple as possible for the reader, these data were not included in the revised version. Should reviewer#2 consider these data need to be included, we are willing to do so.

Figure not included in the manuscript

RT-qPCR analysis of Prune-1 expression in the indicated cell lines and PDXs following pathway inhibition (left) and upon ActivinB stimulation (right).

Figure 4

12/ • Again no mention of the number of biological replica carried out and non quantification of the results with p-values. D458 and D283 show a complete absence of p-SMAD2 in the basal level represented by PBS-treated sample, while in Fig. 2A they seem to express a completely different level of SMAD2 phosphorylation.

p-values, quantifications and number of replicates are added. (see also answer to point 3/Fig2A for the basal level of P-Smad2 (PBS)).

Figure 5

13/ • As above, number of biological replica, quantification and p-values missing.

p-values, quantifications and number of replicates are added.

Figure 6

14/ • As above, number of biological replica, quantification and p-values missing.

p-values, quantifications and number of replicates are added.

15/ • Fig 6C: Authors should show the correlation coefficients for each different subgroups to be able to understand if PMEPA1/INHBB correlation is really important just for GR3 and to justify their sentence "PMEPA1 is the top correlated gene with INHBB within Group 3 MB, showing that their expression is strongly linked in this group. In all MB samples, the correlation is lower than within Group 3 samples".

The *PMEPA1/INHBB* correlation is indeed higher in G3 as compared to all the other groups. See the scatterplots below showing the expression of *INHBB* and *PMEPA1* x- and y-axis respectively. The rho and p-values are the following:

WNT: $r=0.23$ $p=0.056$

SHH: $r=0.147$ $p=0.028$

G3: $r=0.644$ $p=0$

G4: $r=0.508$ $p=0$

The G3 shows the best correlation factor over 0.6 and indeed, the correlation is lower in all other groups. This is now added in a novel Appendix Figure S2.

Appendix Figure S2:

Correlation between *INHBB* (x-axis) and *PMEPA1* (y-axis) in individual groups (blue WNT, red SHH, yellow G3 and green G4, ρ and p-values are indicated on top).

Moreover, it is important to emphasize that the overall level of both *PMEPA1* and *INHBB* is higher in G3 as shown in the scattered plot (presented below and in Figure 4C)

Figure 4C

Correlation between *INHBB* (x-axis) and *PMEPA1* (y-axis) in all groups (individual tumors are labeled according to their groups: blue WNT, red SHH, yellow G3 and green G4, ρ and p-values are indicated on top). This analysis shows that G3 tumors have the highest level of *INHBB* and *PMEPA1*.

This is also confirmed in the following boxplots (Figure 1B (for *INHBB*, Figure 5B for *PMEPA1*).

Figure 1B

Figure 5B

16/• Fig 6H to 6K: Authors should reconstitute *PMEPA1* expression in siRNA-treated cells to show a rescue of the phenotypes presented and exclude off target effects of the siRNA.

Such experiment is very difficult to perform for several reasons. MB, and in particular G3 cell lines, are very fragile and highly sensitive to multiple rounds of transfection and infection. Moreover, several *PMEPA1* isoforms have been described with different biological activities. To overcome this difficulty and to bring evidence that the observed effects are indeed due to "on target" effect we

have repeated our experiments with different individual siRNA that show the same effects. This is included in Figure EV3B-E

Figure EV3B-E

17/ • The authors suggest that "PMEPA1 is an important mediator of Activin signaling-mediated proliferation in Group 3 MB". PMEPA1 protein expression level needs to be checked after ActivinB stimulation and inhibition (blocking antibody) for the claim to be justified.

As requested by the reviewer, these experiments were performed and are included in the revised manuscript in new Figure 5G (in cell lines at the protein level, quantification in Appendix Figure S2D), Appendix Figure S2C-D (in cell lines at RNA level), Figure 6E (in PDXs at the protein level, quantification in Appendix Figure S3C) Appendix Figure S3B (in PDXs at the RNA level). Moreover and as requested by reviewer#3, treatment with Follistatin, a natural inhibitor, was also included. All these data were quantified and p-values are provided. Experiments were reproduced at least 3 time independently.

Figure 5G (in cell lines at the protein, quantification in Appendix Figure S2D)

Appendix Figure S2C (in cell lines at the RNA level)

Figure 6 (in PDXs at the protein level, quantification in Appendix Figure S3C)

Appendix Figure S3B (in PDXs at the RNA level)

18/ • It would be extremely interesting to see the level of PMEPA1 in patients analysed in Fig.1A to strengthen the correlation proposed by the authors.

As requested, we investigated the level of PMEPA1 protein in patient samples by WB and observed a very high correspondence with the level of P-Smad2 (data included in new Figure 5D-F and below). We also showed their high correlation in MB and in the different groups (Appendix Figure S2B)

Figure 5D

This result further supports the fact that, as discussed above, a high level of P-Smad2/β-actin is more relevant than P-Smad2/total Smad2 to evidence pathway activation. Indeed, we also included quantification showing that PMEPA1 level is higher in G3 (Figure 5E) and it is highly correlated with the relative level of p-Smad2 (Figure 5F). A very high correlation factor was found across groups ($r=0.887$ and a p -values $< 0,0001$) and in the different groups with the highest correlation score in G3 (Appendix Figure S2B). This is in line with the fact that *PMEPA1* is a well-established Smad2 target gene and also argues that the overall level of P-Smad2 (*ie* normalization P-Smad2/Actin) is a good readout of Smad2 activation.

Figure 5 E

Figure 5F

Figure 7 and Figure S2

19/ • As above, number of biological replica, quantification and p-values missing. p-values, quantification and number of replicates are added.

20/ • Fig.7A: The authors report that they "found significant but heterogeneous levels of P-Smad2 in the three Group 3 PDX tested". Is not clear how they can define these levels as significant since there isn't any quantification or p-value. Moreover they should show non-GR3 PDX as a control to show whether there is a GR3-specific increased phosphorylation of SMAD2 also in the PDX model. Our previous statement was misleading. We did not mean significant as significant but as important/high. We have modified our statement. We now refer to "As observed in Group 3 patient samples and cell lines, we found heterogeneous levels of P-Smad2, from high to moderate, in the three Group 3 PDXs tested (Figure 6A). PDX4 displayed a very strong activation of the pathway, similar to that observed in the 1603MED cell line.". PDX4 displays a strong P-Smad2 signal, comparable with that observed in 1603MED cell line (see Figure 6A and below). Quantifications were now included. We also measured P-Smad2 levels in a SHH PDX and did not detect any P-Smad2. This figure is provided to the reviewer below. Since this study is too limited (only one PDX), we have not included these data in the revised manuscript but we are willing to do so if the reviewer thinks that it should.

Figure provided to the reviewer

21/ • Fig. 7G: Protein levels of PMEPA1, OTX2 and MYC after the treatments must be showed. As requested, protein levels of PMEPA1, OTX2 and MYC after the treatments are now included in the revised version of the manuscript (new Figure 6E and quantification in Appendix Figure S3C)

Figure 6E

22/ • Fig. 7H: The authors should show pSMAD2, PMEPA1, ACTIVIN B expression in tumour after the treatment with Galunisertib, which is not clear if it's a specific inhibitor of Activin B. We have now included IHC for pSMAD2 in tumors treated or not with Galunisertib (Fig 7C). Unfortunately, we have not been able to set up appropriate IHC conditions for ActivinB and PMEPA1 staining due to the lack of suitable antibodies. A decrease in p-Smad2 was observed after Galunisertib treatment (Figure 7C). Regarding specificity of Galunisertib and all the other inhibitors, we performed experiments to demonstrate that they are able to inhibit ActivinB-induced Smad2 phosphorylation (Appendix Figure S1).

Appendix Figure S1

23/• Fig. 7I and 7H: The authors should check the GR3 subtype of the PDXs that they used. This would strengthen the correlation that they are proposing with the alpha subtype.

Dr Cavalli who identified the different subtypes (Cavalli et al., Cancer Cell 2017) tried to address this point. Unfortunately, due to batch effects and the low number of PDXs, it was not possible to use a clustering method. Consequently, we have tempered our conclusions regarding the G3 α subgroup. The title and the abstract have been modified accordingly.

Nevertheless, we used a less rigorous method that is not precise enough to be published but could be however informative. This approach seems to indicate that the PDXs used in this work belong to the G3 α . Moreover and as indicated in the discussion (original and revised version of the manuscript) "*...PDX4, which expresses very strong level of INHBB, also displays very strong Smad2 activation. This PDX is not MYC-amplified and highly expresses the photoreceptor genes (Garancher et al., 2018).*", PDX4 is a non MYC amplified tumor and expresses high level of photoreceptor genes and probably belongs to the G3 α .

24/ • The authors should describe what is presented in figure S2 and show p-values for it.

The description of Figure S2 (Appendix Figure S3A) is now included in the revised version of the manuscript and p-values are shown in Appendix Table S2.

• To state that "This identifies the α subtype as the photoreceptor Group 3, which displays activation of the Smad2 pathway by an ActivinB autocrine mechanism" the authors need to:

See responses to the different points below.

25/ -1) check correlation of NRL-CRX-INHBB in all the subgroups and subtypes. Are they correlated just in GR3 alpha?

We have likely not been clear enough in our former version. We do not expect a correlation between *INHBB* and *NRL* in G3 α since we did not claim that *NRL* regulates *INHBB* expression nor the reverse. We only wanted to emphasize that *INHBB* is highly expressed in this group which is characterized by high *NRL* expression. Accordingly, this observation is illustrated by boxplot (Figure 7D).

Nevertheless, since the three different reviewers think that this statement is not sufficiently sustained by our data, we do not refer anymore to photoreceptor Group 3 (or G3 α) neither in the title nor in the abstract. This is now a point of discussion.

26/ -2) Check the subtype affiliations of the PDX they used and have PDX from different subgroups and subtype to demonstrate that what they describe holds true only in the GR3 alpha PDXs.

See our answer to point 23. Moreover and as mentioned in response to point 25, we do not refer anymore to photoreceptor Group 3 (or G3 α) neither in the title nor in the abstract in the revised version of the manuscript. This is now a point of discussion."

27/ -3) Check in recently published proteomic data whether there is an increased phosphorylation of p-SMAD2 in GR3 tumours and if this is specific of the alpha subtype.

We failed to find these data in the two proteomic datasets recently published (Archer et al., and Forget et al., Cancer Cell 2018). Unfortunately, it is well established that proteomics allow detection of the most abundant proteins and that many proteins are often not retrieved, especially transcription factors. In fact, analysis of both above mentioned datasets failed to retrieve the phospho-peptides containing activating phosphorylation sites (Ser 465/467) of Smad2. This could be due to the composition of the corresponding peptide and its low molecular weight generated by trypsin digestion. Moreover, these proteomic analyses segregated the G3 into only two groups, which do not strictly overlap with the Cavalli's classification. Therefore, it is unclear where the G3 α subgroup stands among them.

Referee #3 (Comments on Novelty/Model System for Author):

The data is novel as the TGF-beta pathway has not been explored mechanistically in medulloblastoma molecular subgroups before. The medical impact for individualized subgroup-specific therapy is probably low as the Activin pathway is in need of more specific inhibitors.

Referee #3 (Remarks for Author):

Morabito et al. are presenting a manuscript in where the TGF-beta/Activin pathway is explored in the most aggressive subgroup of Group 3 medulloblastoma. They suggest that it is rather Activins than TGF-beta proteins that are potential targets for a subtype of Group 3-alpha patients. They further present how autocrine stimulation of secreted Activin B can stimulate cells via PMEPA1 and that downstream Smad2 pathway activity can be suppressed by Activin B blocking antibodies or pharmacological pathways inhibitors in vitro and also in vivo. The data is interesting but as some critical experiments and relevant inhibitors for the Activin pathway are lacking the results do not provide compelling evidence that the Activin pathway is especially warranted for MB subtype 3-alpha.

Main concerns:

1. In Fig1B it is not clear if differences are statistically significant. It is sufficient to highlight the ones that are significant in main Figure and put the rest in a Supplement Figure.

As requested, we have added all the relevant p-values to show that the differences are statistically significant. All the p-values are included in the supplemental Table S1 to S3. As suggested by the reviewers, we only presented relevant boxplots, *ie* *INHBB*, *TGFB1* and *TGFB3*, in the main Figure 1B. The other data were switched into Figure EV1D.

2. Regarding Figure 2B-F + Figure 7B. All the RT-PCR data have been normalized to HDMB03 (probably because it has lowest P-Smad2 of the Gr3 cell lines) - however this is stated nowhere in the figure legends, methods, etc. Further, X-axis should have the same range for all graphs for better comparisons.

This is now stated in the relevant Figure legends and we homogenized all the x-axes as requested.

3. Regarding Figure 2C. Do not agree with the author's statement that *INHBB* is generally higher expressed in Gr3 cell lines than non-Gr3 cell lines. We investigated the expression of different ligands (Figure 2A-F1D and EV1D) and found a higher expression of *INHBB* in the 1603MED and D283 Group 3 cell lines as compared to the others (Figure 1D2C). (only true for lines 1603MED and D283).

We fully agree with the reviewer and we apologize for this statement that was not clear enough. Indeed, we observed heterogeneous levels of P-Smad2 with very high level in 1603MED and intermediate level in D283. Accordingly, we observed High *INHBB* expression in these cell lines. The text has been modified as follows: "*We investigated the expression of different ligands (Figure 1D and EV1D) and found a higher expression of INHBB in the 1603MED and D283 Group 3 cell lines as compared to the others (Figure 1D).*"

4. In Figure 3C, HDMB03 responds to conditioned media from 1603MED by phosphorylating Smad2; but the authors don't follow up on whether TGF-beta signaling activation has functional consequences on the treated cells (e.g. increase in proliferation) and whether this can be prevented by the anti-Activin B antibody.

Compared to all the other G3 cell lines that grow in suspension, HDMB03 cells are adherent and display very low P-Smad2 basal level. They represent the perfect tool suited for conditioned media experiments aiming at demonstrating the secretion of ActivinB by 1603MED cells. Nevertheless, ActivinB stimulation does not induce cell proliferation in this cell line. The exact reason for that is currently unclear. Nevertheless, we noticed that, in contrast to all other responding G3 cell lines, HDMB03 cells do not express *FOXG1*, which has been shown to play an important role to overcome TGFβ/Activin pathway-induced growth inhibition and to promote cell proliferation (Seoane et al., Cell 2004). Accordingly, *FOXG1* is highly expressed in G3 and G4 in patient samples. See below the boxplot provided to the reviewer (data source: <https://hgserver1.amc.nl/cgi-bin/r2/main.cgi>). Although potentially interesting at the mechanistic level, this is out of the scope of the present manuscript.

Boxplot displaying expression of *FOXG1* across groups (data obtained from <https://hgserver1.amc.nl/cgi-bin/r2/main.cgi>)

5. Regarding Figure 3F & 6I. It is impossible to show proliferation with basically one time point (apart from start point); authors could have at least added one additional time point (Day 1); Guess this is also a transient effect or is cell proliferation inhibited after Day 3?

As requested by the reviewer, we added an additional time point (day 3) but as anticipated by the reviewer, the effect of siRNA is transient and growth inhibition did not persist with time (Figure 2F).

Figure 2F

6. In Figure 4, the data for HDMB03 is missing (lowest basal activation of the Activin pathway) - should benefit the most from exogenous Activin B. Could also not find any information how often cells were treated with Activin B. Please clarify this.

Regarding the lack of data for HDMB03 in previous Figure 4, please see our answer to point 4 of this referee. Cells were treated only once with ActivinB. This is now stated in the Materials and Methods.

7. In Figure 6, it is confusing why authors are focus on PMEPA1 at all. What role does it play for the story of the paper (there's no further focus on PMEPA1 in the remaining paper)? Is this protein druggable or downstream of pSmad2? Suggested experiment: treat HDMB03 with Activin B and show upregulation of PMEPA1 and further check for pSmad2.

As mentioned in the manuscript, *PMEPA1* is a well-established target gene of the Smad2 signaling in response to Activin (Coda et al., 2017 Elife 2017) or TGF β (Fournier et al., Cancer Cell, 2015) in different cell types. Thus, *PMEPA1* expression constitutes a readout of the pathway activation and may serve as such. Moreover, we now included data on patient samples showing an excellent correlation between p-Smad2 and PMEPA1 expression (protein level) (Figure 5D and E-F). We strongly believe that this aspect is highly important and relevant since it allows establishing that the observed P-Smad2 has a functional outcome.

At the mechanistic level, we demonstrate that PMEPA1 participates in activin B-induced cell proliferation. We agree with the reviewer that it is likely only a part of the mechanism, PMEPA1 being required but likely not sufficient. If the reviewer feels that this part is unnecessary, we are ready to withdraw this set of functional data, although we still believe that it represents an interesting aspect of the story.

Regarding HDMB03, they do not constitute a good model to perform mechanistic analyses since ActivinB does not promote cell proliferation in this cell line (see also our answer to point 4).

8. A major problem is that the drugs used in the manuscript are all TGFBR1 inhibitors and not specific ACVR inhibitors. Both pathways will stimulate downstream pSmad2 as they show with either TGF-beta or Activin B stimulation. Would thus be important to find an ACVR specific drug (ALK4/7 specific) or interesting to see how e.g. Decorin or FST treatment would affect tumor growth, or use TGF-beta stimulation in combination with available drugs that target ALK4, 7 as well as ALK5. The blocking antibody itself would be interesting to test *in vivo* or valid reasons for not using this should be explained.

In order to perform reliable *in vivo* experiments, good pharmacokinetics as well as an ability to cross the blood brain barrier (BBB) are absolutely required. In general, antibodies do not fulfill these requirements. We used Galunisertib because it is well characterized for all these aspects and, importantly, is currently in clinical trial for brain tumors.

As requested by the reviewer we performed, and included in the revised version of the manuscript, experiments with Follistatin (FST). FST showed very strong inhibiting effects on P-Smad2 in different models (cell lines 1603MED (Figure 2B), and PDX4 (Figure 6E). We also showed that it decreases PMEPA1 protein levels (see Figure 5G and 6E (at the protein level and Appendix Figure S2D and S3B (RNA level). See also below).

Figure 5G (Protein level)

Appendix Figure S2D (RT-qPCR)

Figure 6E (Protein level)

Appendix Figure S3B (RT-qPCR)

9. The TGFBR1 inhibitor Galunisertib is poorly introduced and it is not clear why authors don't show data on this *in vitro* as they do with LY364947 and SB431542? Does Galunisertib have the same effects as LY364947 and SB431542 (e.g. on proliferation, P-Smad2 levels etc.).

We have now better introduced Galunisertib by modifying the text as followed " ...Galunisertib, a pharmacological inhibitor currently in clinical trial for Glioblastoma, Galunisertib is described as a TGF β type I inhibitor but since TGF β and activin type I receptors are very similar, it also inhibits very efficiently ActivinB-induced Smad2 activation (Appendix Figure S1). Accordingly, we have recapitulated the major *in vitro* data obtained with LY364947 and SB431542 with Galunisertib (Figure EV4A-C)." We also added a new Figure showing that Galunisertib efficiently inhibits ActivinB-mediated Smad2 activation (Appendix Figure S1).

Appendix Figure S1

As requested by this referee, we repeated the key *in vitro* experiments with Galunisertib and showed that indeed this compound has the same effects as LY364947 and SB431542. These data are now included in the revised version of the manuscript as supplemental Figures (see Figure EV4A-B and below).

Figure EV4A-B

10. In the figure it should also be clearly shown that the treatment was stopped during weekends (no continuous treatment).

We performed a new experiment by modifying the treatment protocol. Mice have been treated twice a day, 7 days a week. This is included in the Figure and clearly stated in the legends. (see also point 12)

11. In Figure 7H, it is strange that there's no *in vitro* data for PDX7 (used for *in vivo* experiment), only for PDX3 and PDX4.

PDX4, which shows the strongest P-Smad2 signal, was used for *in vivo* experiments and not PDX7. PDX7 was omitted in our previous analysis since it displays low P-Smad2 levels. Indeed, we

thought that assessing the effect of inhibitors on potential target genes was not informative in a cell line that does not display strong pathway activation. To meet the referee's concern, we nevertheless included data for PDX7 in the new Figure 6E (at the protein level) and Appendix Figure S3B (see below).

Figure 6E

Appendix Figure S3B

12. Would mice benefit from long-term treatment? How does the Galunisertib treatment perform in comparison to standard treatment options (e.g. Cisplatin)? What are the actual effects on the tumour? Smaller? More/less apoptosis? Less proliferation/cell cycle arrest/differentiation? What about combination treatments? From the in vitro data I assume tumor growth is slowed down, but tumour cells are not killed (a second drug could perhaps do the job).

We tested longer treatment but did not observe major differences. As mentioned in response to point 10 from this referee, we have now modified the treatment protocol by treating mice twice a day, every day. This novel protocol together with combination with cisplatin is now included in Figure 7A. We observed a longer survival with this type of treatment (see below) as compared to the previous one (1xday and 5 days per week, see previous version of the manuscript).

As requested, mice were treated with Cisplatin (3 times as performed by Dr Marino's group in Niklison-Chirou et al., Genes and Dev, 2017) or with the combo (Cisplatin/Galunisertib). We observed that Galunisertib is highly efficient although the treatment protocol was completely different and cannot be compared. We did not observe any gain with the Cisplatin/Galunisertib combination (See below and Figure 7A) in terms of survival. We have currently no explanation for this result. More work is needed to identify good combinations with Galunisertib. We observed that indeed the tumors are smaller (Figure 7B), with a decrease of P-Smad2 by IHC (Figure 7C) and in proliferation (KI67) (Figure EV4D) when Galunisertib is included in the treatment (see also below).

Figure 7A-B

Figure 7C (IHC P-Smad2)

Figure EV4D (IHC KI67)

13. In Figure 7I-J, I don't think authors can prove that Activin pathway inhibition is especially warranted for subtype 3alpha although the correlations to NRL and CRX exist. What if you treat a SHH PDX with Galunisertib? Is the drug really specific to Group 3? They only have in vivo results from one Gr. 3 PDX and the cell lines do not respond to Activin/TGF signalling perturbations corresponding following their MYC, p53 or INHBB status. Authors don't know why some Gr3 have higher basal TGFbeta activity than others.

We agree with the reviewer that Galunisertib could also be efficient in other patients than those belonging to photoreceptor G3 α . Actually and as mentioned in our previous version, published data showed that some SHH patients display Smad2 activation (Aref et al., Brain Pathol 2013) as well as in SHH animal model (Gate et al., PNAS 2014). As mentioned by the reviewer, most of the G3 cell lines are indeed MYC amplified including 1603MED. In agreement with these reports, we did not claim that only G3 α tumors could respond to Galunisertib but we underlined that the G3 α subtype is particularly enriched in tumors displaying pathway activation through an autocrine loop. Indeed, we stated in the discussion of the previous version of the manuscript "... *It should be nevertheless mentioned that the 1603MED cell line is also characterized by high INHBB expression and high Smad2 activation but is MYC amplified and does not express high level of photoreceptor genes (Raso et al., 2008). Thus, we proposed that activation of the Smad2/3 pathway involving an ActivinB autocrine stimulation, is enriched in subtype Group 3 α , although not completely limited to this subtype...*"

We still believe that taken together our results suggest that G3 α is enriched in high P-Smad2 tumors. This claim is based on the highest expression level of ActivinB (*INHBB*) and *PMEPA1* in G3 α and their high level of correlation. Since we showed that the level of *PMEPA1* is a readout of pathway

activation, this supports that the Smad2 pathway is activated by an autocrine loop in G3 α and is particularly high in this subtype. Nevertheless, since the 3 reviewers appear somehow confused by this claim, we have modified the title and the abstract and do not refer to this point. This is only included as a point of discussion in the revised version of the manuscript.

Minor concerns:

1. There are a few spelling errors/grammar mistakes:

End of abstract text: orthotopically

Introduction: photoreceptor genes in which few amplifications...

A few sentences in the first section of page 4 are oddly written. The description of the subtypes on page 5 should be clearer and better written with regard to the 3 distinct Gr3 subtypes in the Cavalli set.

Page 9: ...decreased of cell growth (Figure 3F).

Material/method: Dr. Milde is misspelled

Figure 7 legend: Kaplan-Meier misspelled

We thank the reviewer and these have been modified accordingly.

2. Put Figure 7G in a Supplementary figure.

This has been modified accordingly

3. In Figure 4F: Cells stop proliferating at day 4 (plateau); interesting to see what happens if you keep adding Activin.

Activin is added only once in these experiments. When D283 cells reach a plateau, they start to die. It is thus difficult to perform this experiment.

4. In Figure 6K. Why do control D283 cells stop proliferating after 80 hrs? Cannot be lack of space, because only 30% confluent according to Incucyte.

D283 cells never reach confluency. They are semi-adherent cells. They stop growing at one point, reach a plateau and start to die.

2nd Editorial Decision

17 May 2019

Thank you for the submission of your revised manuscript to EMBO Molecular Medicine.

As you will see from the enclosed reports, the referees appreciate (as we do) the work provided during the revisions in order to improve the manuscript and answer the reviewers' points. However, both referees #2 and #3 still have concerns regarding the quantification and analysis of the western blots (mostly in figure 1). Addressing these concerns will be necessary for acceptance of the manuscript. However, I would like to stress that at that stage, we mostly ask for quantification and re-writing, and not for you to provide extensive additional experiments.

***** Reviewer's comments *****

Referee #1 (Remarks for Author):

No more questions

Referee #2 (Remarks for Author):

The authors have adequately addressed many of the points I have raised. However, one major point remains a concern. The quantification of the WB presented in Figure 1A continue to be not convincing; yet this is a key finding driving all the subsequent experiments described in this manuscript. It is essential it is appropriately addressed.

1) The authors explain that the normalization asked by the reviewers (i.e. normalize on the loading control and then on the total SMAD protein) is not biologically relevant since they want to analyse the active nuclear fraction of p-SMAD2. Why then would they normalize to a cytoplasmic protein, namely b-actin? Why are they not using a nuclear protein as loading control? If they want to focus on p-SMAD active in the nucleus, either they analyse nuclear extracts or they normalise to the correct control.

2) Attention is also required with the densitometric quantification. What are they quantifying in samples such as the #351? Or #371, 4, 5, 40 and 393. I do not see a band for p-SMAD in these samples. This data analysis does not look accurate.

3) Finally, if as the authors claim normalization p-SMAD/TOT SMAD is not biologically significant, why then this precise quantification is used for Fig 1C and all the following figures?

Referee #3 (Comments on Novelty/Model System for Author):

I think they used adequate patient material, model systems in vitro and in vivo, and further used drugs that are justified to be specific for their intended TGF-beta/Activin inhibition.

Referee #3 (Remarks for Author):

Morabito et al. show how autocrine ActivinB mechanisms promote TGF-beta/Activin signalling in Gr. 3 MB. The revised manuscript is much improved with more samples, experiments and convincing data. I am also happy that authors toned down that Activin pathway inhibition/Galunisertib therapy was especially warranted for Group 3 alpha patients and for modifying their title and abstract accordingly. Still, I would have a few points/concerns to address in an additional revision:

1. Regarding Figure 1A: P-Smad2 levels in some tumours are quite different from the WB in the first version of the manuscript. See for example patients #4 or #40 (SHH tumours). Shouldn't the tumour samples have more consistent p-Smad2 levels as compared to the cell lines?

2. General concern: If it is so important to use p-Smad2/Actin levels (instead of Smad2/total Smad2 levels) in the tumour samples (see reviewer 1, Fig1), why aren't the authors using it throughout the manuscript. They are instead using p-Smad2/total Smad2 levels in the rest of the paper - see 1C, 2A-2C, 2E, 3A, 3E, 4A, 4E.

3. Regarding Figure 3: According to comments, HDMB03 which grows adherently and display very low p-Smad2 levels, does not increase proliferation in response to ActivinB stimulation. This cell line should be included in Fig 3, otherwise it looks like all Grp3 cell lines respond to ActivinB stimulation in the same way, which is not true. Or at least, they should state it in the text.

4. Regarding Figure 5D: Even though they state it in the figure legend, it feels a bit strange to display the exact same WB for pSmad2/Smad2 as in Fig 1A (except that they include PMEPA1). PMEPA1 WB looks bad, especially in Grp 3 where it is too much exposed! A less exposed staining would be preferable.

5. Regarding Figure 7C: Quantification of p-Smad2 should be relevant but is missing.

Minor comments:

1. Authors comment on using longer exposure times of some of the WBs in the new version. However, why does the Actin look weaker in many of the WBs?

2. Regarding Figure EV4: Quantification of Ki67 and CC3 is missing. Cleaved caspase 3 is further misspelled here.

3. Comment regarding Figure 7: Galunisertib treatment prolongs survival and decreases tumour size, whereas combination treatment with Cisplatin is not more effective. Perhaps useful to first shrink

the tumours with Galunisertib and then hit it with Cisplatin or irradiation that is probably even better at reaching to the brain tumour than cisplatin?

4. Regarding Figure 7D: There is a minor mistake with the colours for the different subgroups in the legend.

2nd Revision - authors' response

28 June 2019

Referee #1 (Remarks for Author):

No more questions

Referee #2 (Remarks for Author):

The authors have adequately addressed many of the points I have raised. However, one major point remains a concern. The quantification of the WB presented in Figure 1A continue to be not convincing; yet this is a key finding driving all the subsequent experiments described in this manuscript. It is essential it is appropriately addressed.

1) The authors explain that the normalization asked by the reviewers (i.e. normalize on the loading control and then on the total SMAD protein) is not biologically relevant since they want to analyse the active nuclear fraction of p-SMAD2. Why than would they normalize to a cytoplasmic protein, namely b-actin? Why are they not using a nuclear protein as loading control? If they want to focus on p-SMAD active in the nucleus, either they analyse nuclear extracts or they normalise to the correct control.

The western blot experiments presented in this figure were performed on patient samples, which is a highly valuable and scarce material for this rare tumor. Only total cell extracts were prepared and not nuclear extracts. Indeed, nuclear extracts are very difficult to prepare from small amounts of frozen tissue and are not currently available. Total cell extracts are commonly normalized with β -actin or GAPDH even when the level of a nuclear protein is investigated. Indeed, the main characteristic of a good normalizing protein is 1/ to be present and easily detectable 2/ to display a level that remains unchanged in most conditions tested (i.e. not regulated). Its subcellular localization does not matter since it is supposed to reflect the amount of total proteins present in the different samples. Accordingly, nuclear proteins are very rarely used for total cell extracts normalization in publications. Owing to the characteristics listed above and as we did, the most frequent normalizers used for total cell extracts are b-actin, tubulin or GAPDH, whatever the subcellular localization of the investigated protein. Of note, we have checked some samples with GAPDH and our normalization remains mostly unchanged.

2) Attention is also required with the densitometric quantification. What are they quantifying in samples such as the #351? Or #371, 4, 5, 40 and 393. I do not see a band for p-SMAD in these samples. This data analysis does not look accurate.

For each figure, many different exposures were generated, but the one displayed was chosen to have no saturating signal for a band of interest throughout the blot lanes. Since the signal was captured by a CCD camera, it is possible however to quantify band densitometry from raw data, even when it is barely visible on the exposure chosen for the final figure. We provide below a longer exposure of the P-Smad2 blot, which shows the presence of the weak P-SMAD bands.

3) Finally, if as the authors claim normalization p-SMAD/TOT SMAD is not biologically significant, why then this precise quantification is used for Fig 1C and all the following figures? The p-SMAD/TOT SMAD normalization was requested by the reviewers. To fulfill this request, we provided this type of normalization for all figures. However, we fully agree with the reviewer that the p-SMAD/ β -Actin normalization should be provided for better consistency. We have now replaced this quantification/normalization in all figures and provided, for reader information, all the normalizations to total Smad in the new supplemental Appendix figure S5.

Referee #3 (Remarks for Author):

Morabito et al. show how autocrine ActivinB mechanisms promote TGF-beta/Activin signalling in Gr. 3 MB. The revised manuscript is much improved with more samples, experiments and convincing data. I am also happy that authors toned down that Activin pathway inhibition/Galunisertib therapy was especially warranted for Group 3 alpha patients and for modifying their title and abstract accordingly. Still, I would have a few points/concerns to address in an additional revision:

1. Regarding Figure 1A: P-Smad2 levels in some tumours are quite different from the WB in the first version of the manuscript. See for example patients #4 or #40 (SHH tumours). Shouldn't the tumour samples have more consistent p-Smad2 levels as compared to the cell lines?

We agree with the reviewer that some variability can be observed for few SHH tumors. However, tumor material is likely more variable than cell lines, since it is well established that tumors can be very heterogeneous. We believe that this intratumoral heterogeneity may be the source of this observed variability. Such differences were not observed in G3 tumors and, importantly, this variability did not modify our statement that G3 tumors have high p-Smad2 level with some intertumoral heterogeneity.

2. General concern: If it is so important to use p-Smad2/Actin levels (instead of Smad2/total Smad2 levels) in the tumour samples (see reviewer 1, Fig1), why aren't the authors using it throughout the manuscript. They are instead using p-Smad2/total Smad2 levels in the rest of the paper - see 1C, 2A-2C, 2E, 3A, 3E, 4A, 4E.

Copy of response to point 3 of Reviewer #2:

The p-SMAD/TOT SMAD normalization was requested by the reviewers. To fulfill this request, we provided this type of normalization for all figures. However, we fully agree with the reviewer that the p-SMAD/ β -Actin normalization should be provided for better consistency. We have now replaced this quantification/normalization in all figures and provided, for reader information, all the normalizations to total Smad in the new supplemental Appendix figure S5.

3. Regarding Figure 3: According to comments, HDMB03 which grows adherently and display very low p-Smad2 levels, does not increase proliferation in response to ActivinB stimulation. This cell line should be included in Fig 3, otherwise it looks like all Grp3 cell lines respond to AcitivinB stimulation in the same way, which is not true. Or at least, they should state it in the text.

We have modified the text to state that ActivinB does not increase the proliferation rate in HDMB03. "*It remains to be determined why Activin B did not promote cell growth while activating the pathway in HDMB03.*" The results are displayed in Supplemental Appendix Figure S2.

4. Regarding Figure 5D: Even though they state it in the figure legend, it feels a bit strange to display the exact same WB for pSmad2/Smad2 as in Fig 1A (except that they include PMEPA1). PMEPA1 WB looks bad, especially in Grp 3 where it is too much exposed! A less exposed staining would be preferable.

We agree with the referee. We now include a short and a long exposure of the PMEPA1 panel and the Figure 5D is switched to Figure 1A as requested.

5. Regarding Figure 7C: Quantification of p-Smad2 should be relevant but is missing.

We have now included the quantification of these IHC (Figure 7B)

Minor comments:

1. Authors comment on using longer exposure times of some of the WBs in the new version. However, why does the Actin look weaker in many of the WBs?

The longer exposure was shown only for the p-Smad2 WB, not for the β -actin WB whose panels were unchanged. The exposure is independently chosen for each protein in order to avoid saturating signal. Note that one cannot compare the signal intensity for two distinct proteins using different antibodies.

2. Regarding Figure EV4: Quantification of Ki67 and CC3 is missing. Cleaved caspase 3 is further misspelled here.

Quantifications are now added in new figure EV4 panel 1, as requested. We did not detect any significant change in Ki67 and Cleaved caspase 3 staining. This is stated in the text.

3. Comment regarding Figure 7: Galunisertib treatment prolongs survival and decreases tumour size, whereas combination treatment with Cisplatin is not more effective. Perhaps useful to first shrink the tumours with Galunisertib and then hit it with Cisplatin or irradiation that is probably even better at reaching to the brain tumour than cisplatin?

We agree with the referee that the timing of combination might be not optimal or that a combo with irradiation might be more efficient. We include that as a point of discussion in the result section: "We did not observe any benefit from the combination of Galunisertib with Cisplatin" modified by *"Although, we did not observe any benefit from the combination of Galunisertib with Cisplatin (Figure 7A-C and EV4D-E), we cannot not exclude that different treatment kinetics could be more efficient. In this respect, other combinations with different drugs or radiotherapy remain to be evaluated."*

4. Regarding Figure 7D: There is a minor mistake with the colours for the different subgroups in the legend.

We thank the reviewer for pointing out this mistake. This has been modified accordingly.

Corresponding Author Name: Celio Pouponnot
 Journal Submitted to: EMBO Molecular Medicine
 Manuscript Number: EMM-2018-09830